# Deep learning empowered volume delineation of whole-body organs-at-risk for accelerated radiotherapy

Feng Shi[1,7], Weigang Hu[2,3,7], Jiaojiao Wu[1,7], Miaofei Han[1], Jiazhou Wang [2,3], Wei Zhang[4], Qing Zhou[1], Jingjie Zhou[4], Ying Wei[1], Ying Shao[1], Yanbo Chen[1], Yue Yu[1], Xiaohuan Cao[1], Yiqiang Zhan[1], Xiang Sean Zhou[1], Yaozong Gao[1] ✉ & Dinggang Shen [5,1,6] ✉

In radiotherapy for cancer patients, an indispensable process is to delineate organs-at-risk (OARs) and tumors. However, it is the most time-consuming step as manual delineation is always required from radiation oncologists. Herein, we propose a lightweight deep learning framework for radiotherapy treatment planning (RTP), named RTP-Net, to promote an automatic, rapid, and precise initialization of whole-body OARs and tumors. Briefly, the framework implements a cascade coarse-to-fine segmentation, with adaptive module for both small and large organs, and attention mechanisms for organs and boundaries. Our experiments show three merits: 1) Extensively evaluates on 67 delineation tasks on a large-scale dataset of 28,581 cases; 2) Demonstrates comparable or superior accuracy with an average Dice of 0.95; 3) Achieves near real-time delineation in most tasks with <2 s. This framework could be utilized to accelerate the contouring process in the All-in-One radiotherapy scheme, and thus greatly shorten the turnaround time of patients.

Cancer is considered to be a major burden of disease with rapidly increasing morbidity and mortality worldwide[1–3]. It is estimated to be 28.4 million new cancer cases in 2040, a 47.2% rise from the corresponding 19.3 million new cancer cases that occurred in 2020. Radiotherapy (RT) is used as the fundamentally curative or palliative treatment for cancer, with approximately 50% of cancer patients receiving benefits from RT[4–6]. Considering that high-energy radiation can damage genetic materials of both cancer and normal cells, it is important to balance the efficacy and the safety of RT, which highly depends on the dose distribution of irradiation, as well as the functional status of organs-at-risk (OARs)[6–9]. Accurate delineation of tumors and OARs can directly influence outcomes of RT, since

inaccurate delineation may lead to overdosing or under-dosing issues and increase the risk of toxicities or decrease the efficacy of tumors. Therefore, in order to deliver a designated dose to the target tumor while protecting the OARs, accurate segmentation is highly desired.

The routinely clinical RT workflow can be divided into four steps, including (1) CT image acquisition and initial diagnosis, (2) radiotherapy treatment planning (RTP), (3) delivery of radiation, and (4) follow-up care. This is guided by a team of healthcare professionals, such as radiation oncologists, medical dosimetrists, radiation therapists, and so on[10,11]. Generally, during the RTP stage, the contouring of OARs and target tumors is performed manually by radiation oncologists and dosimetrists. Note that the reproducibility and consistency of

[1]Department of Research and Development, Shanghai United Imaging Intelligence Co., Ltd., Shanghai, China. [2]Department of Radiation Oncology, Fudan University Shanghai Cancer Center, Shanghai, China. [3]Department of Oncology, Shanghai Medical College, Fudan University, Shanghai, China. [4]Radiotherapy Business Unit, Shanghai United Imaging Healthcare Co., Ltd., Shanghai, China. [5]School of Biomedical Engineering, ShanghaiTech University, Shanghai, China. [6]Shanghai Clinical Research and Trial Center, Shanghai, China. [7]These authors contributed equally: Feng Shi, Weigang Hu, Jiaojiao Wu. ✉e-mail: yaozong.gao@uii-ai.com; Dingang.Shen@gmail.com

manual segmentation are challenging due to intra- and inter-observer variability[12]. Also, manual process is very time-consuming, and often takes hours or even days per patient, leading to significant delays in RT treatment[12,13]. Therefore, it is desired to develop fast segmentation approach to achieve accurate and consistent delineation for both OARs and target tumors.

Most recently, deep learning-based segmentation has shown enormous potential in providing accurate and consistent results[10,11,14–16], in comparison to most classification and regression approaches, such as atlas-based contouring, statistical shape modeling, and so on[17–20]. The most popular architecture is convolutional neural networks (CNNs)[21–23], including U-Net[24,25], V-Net[26], as well as nnU-Net[27], which achieve excellent performance in Medical Image Decathlon Segmentation Competition. Besides, other hybrid algorithms also have shown outstanding segmentation performance[28–30], i.e., Swin UNETR[31]. However, deep learning-based algorithm needs specific computing resources such as graphics processing unit (GPU) memory, especially for 3D image processing[13], thus leading to limited clinical applications in practice.

To address the above challenges, herein, we propose a lightweight automatic segmentation framework, named RTP-Net, to greatly reduce the processing time of contouring OARs and target tumors, while achieving comparable or better performance with the state-of-the-art methods. Note that this framework has potential to be used in the recent emerging All-in-One RT scheme (Fig. 1). All-in-One RT intends for providing a one-stop service for patients by integrating the CT scanning, contouring, dosimetric planning, and image-guided in situ beam delivery in one visit. In this process, the contouring step could be accelerated by the artificial intelligence (AI) algorithm from hours to seconds, followed by an oncologist's review with minimal required modifications, which can significantly improve efficiency and accelerate process at the planning stage (Fig. 1a). With the development of the RT-linac platform and the integration of multi-functional modules (i.e., fast contouring, auto-planning, and radiation delivery), the All-in-One RT can shorten the whole RT process from days to minutes[32] (Fig. 1b).

## Results and discussion
### RTP-Net for efficient contouring of OARs and tumors

To increase accuracy and also save time for RTP, we propose a light-weight deep learning-based segmentation framework, named as RTP-Net, as shown in Fig. 2, for automated contouring of OARs and tumors. In particular, three strategies are designed to (1) produce customized segmentation for given OARs, (2) reduce GPU memory cost, and (3) also achieve rapid and accurate segmentation, as briefed below.

(1) Coarse-to-fine strategy. This is proposed for fast segmentation of 3D images by using a coarse-resolution model to localize a minimal region of interest (ROI) that includes the to-be-segmented region in the original image, and then using a fine-resolution model to use this ROI as input to obtain detailed boundaries of the region (Fig. 2a). This two-stage approach can effectively exclude a large amount of irrelevant information, reduce false positives, and improve segmentation accuracy. At the same time, it helps reduce GPU memory cost and improve efficiency of segmentation. We adopt VB-Net here, as proposed in our previous work[33], to achieve quick and precise segmentation. It is developed based on the classic V-Net architecture, i.e., an encoder-decoder network with skip connection and residual connection, and further improved by adding the bottleneck layer. The VB-Net has achieved first place in the SegTHOR Challenge 2019 (Segmentation of Thoracic Organs at Risk in CT Images). The detailed architecture and network settings can be obtained in Methods and Table 1.

(2) Adaptive input module. To segment both small and large ROIs, an adaptive input module is also designed in VB-Net architecture, by adding one down-sampling layer and one up-sampling layer to the beginning and the end of the VB-Net, respectively, according to the size of the target ROI (Fig. 2b). Both resampling operations are implemented through a convolution layer, which can learn best parameters among processes and reduce GPU memory simultaneously.

(3) Attention mechanisms. For accurate delineation of the target volume (PTV/CTV), two attention mechanisms are particularly

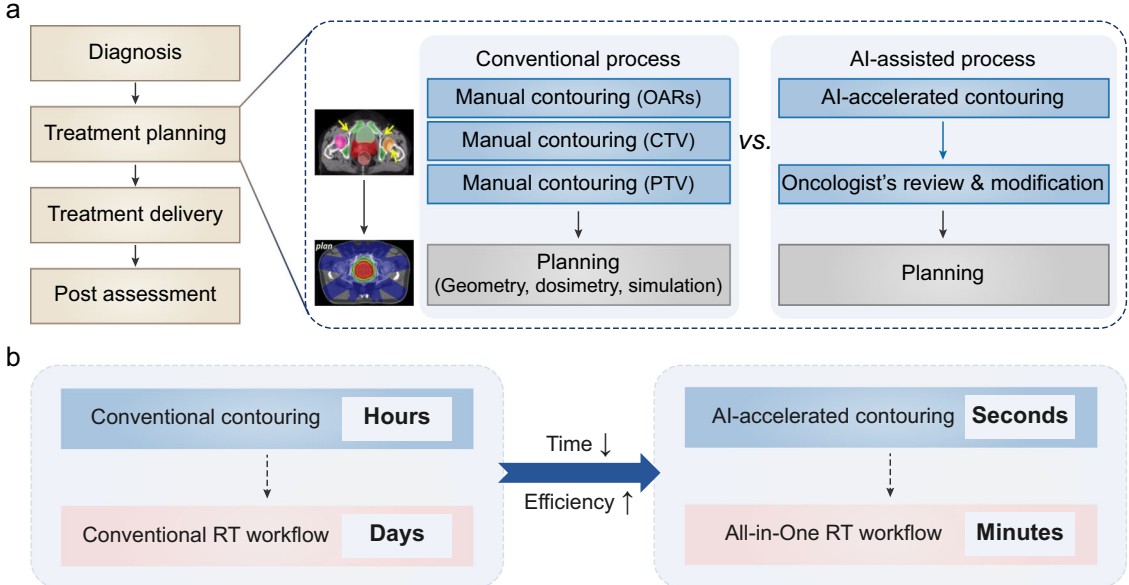

**Fig. 1 | Artificial intelligence (AI)-accelerated contouring promotes All-in-One radiotherapy (RT). a** The process overview of conventional RT *vs.* AI-accelerated All-in-One RT. The RT workflow can be divided into four steps, in which treatment planning step can be accelerated by AI. Conventional treatment planning includes manual contouring of organs-at-risk (OARs), clinical target volume (CTV), and planning target volume (PTV), followed by the planning procedures. The contouring step can be accelerated by AI algorithms, followed by an oncologist's review with minimal required modification. **b** The time scales of contouring and RT workflow in the conventional RT and the AI-accelerated All-in-One RT, respectively. The contouring step can be accelerated by AI from hours to seconds, and the whole RT process can be shortened from days to minutes.

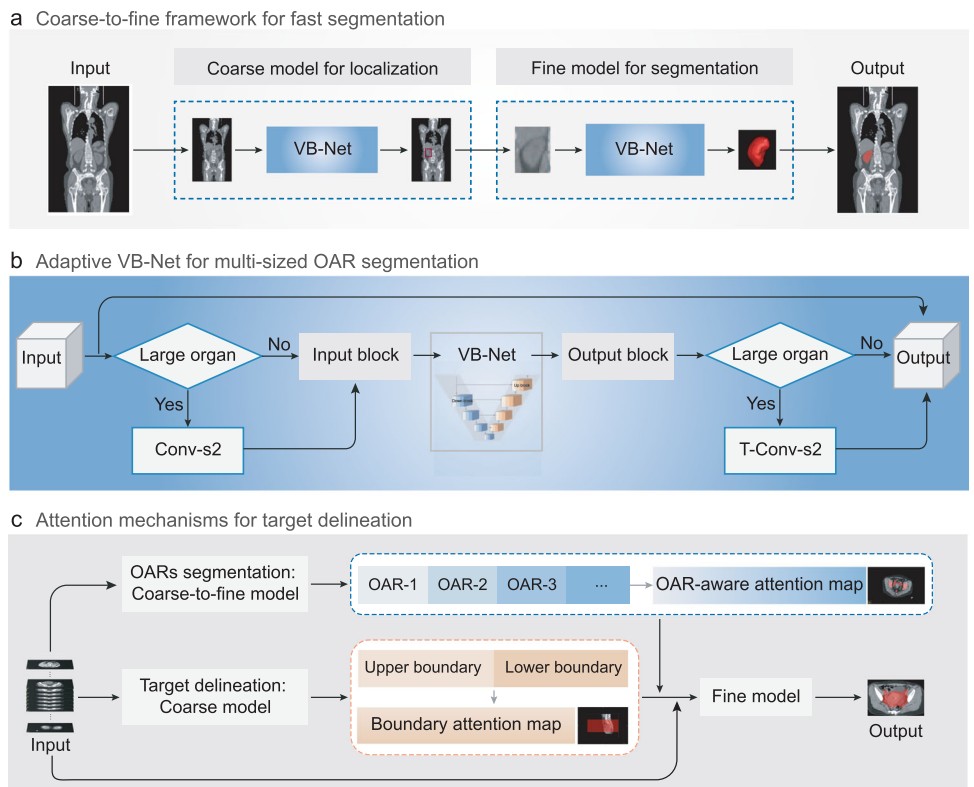

**Fig. 2 | Schematic representations of RTP-Net for fast and accurate delineation of organs-at-risk (OARs) and tumors. a** Coarse-to-fine framework with multi-resolutions for fast segmentation. A coarse-resolution model is to localize the region of interest (ROI) in the original image (labeled in the red box), and a fine-resolution model is to refine the detailed boundaries of ROI. **b** Adaptive VB-Net for multi-sized OAR segmentation, which can be also applied to large organs. This is achieved by adding a stridden convolution layer with a stride of 2 (Conv-s2) and a transposed convolution layer with a stride of 2 (T-Conv-s2) to the beginning and the end of the VB-Net, respectively. **c** Attention mechanisms used in the segmentation framework for accurate target volume delineation. The OAR-aware attention map is generated by the fine-level OAR segmentation, and the boundary-aware attention map is generated by the coarse-level target volume bounding box. Two attention maps combined with multi-dimensional adaptive loss function are adopted to modify the fine-level model for obtaining accurate target delineation.

developed, i.e., the OAR-aware attention map and the boundary-aware attention map (Fig. 2c). The OAR-aware attention map is generated by the fine-level OAR segmentation, while the boundary-aware attention map is applied in the coarse-level PTV/CTV bounding box. The OAR-aware attention map is utilized as an additional constraint to improve the performance of the fine-resolution model. Specifically, the input of the fine-resolution model is the concatenation of the raw image with its OAR-aware attention map and boundary-aware attention map in a channel-wise dimension. That is, both attention mechanisms (combined with the multi-dimensional adaptive loss function) are adopted to modify the fine-level VB-Net.

In summary, the proposed RTP-Net framework can segment target volumes as well as multiple OARs in an automatic, accurate, and efficient manner, which can be then followed by in-situ dosimetric planning and radiation therapy to eventually achieve All-in-One RT. In our developed segmentation framework, a set of parameters are open for users to adjust, including pre-processing configuration, training strategy configuration, network architecture, and image inference configuration. Also, considering the diversity of different imaging datasets, such as imaging modality, reconstruction kernels, image spacing, and so on, the users are allowed to customize a suitable training configuration setting for each specific task. The recommended configuration setting of our multi-resolution segmentation framework is summarized in Table 1 for reference.

## Evaluation of segmentation results for whole-body OARs

Segmentation performance of the proposed RTP-Net is extensively evaluated on the whole-body organs, including overall 65 OARs distributed in the head, chest, abdomen, pelvic cavity, and whole body, in terms of both accuracy and efficiency. Importantly, a large-scale dataset of 28,219 cases is experimented, of which 4,833 cases are used as the testing set (~17%) and the remaining cases serve as the training set (Supplementary Fig. 1).

The accuracy of the segmentation is quantified by the Dice coefficient, ranging from 0 to 1, with Dice coefficient of 1 representing perfect overlapping between the segmented result and its ground truth. As shown in Fig. 3 and Supplementary Table 1, the Dice coefficients of automatic segmentations on a set of OARs are measured. Totally, we implement 65 segmentation tasks, including 27 OARs in the head part, 16 OARs in the chest part, 10 OARs in the abdomen part, 9 OARs in the pelvic cavity part, and 3 OARs in the whole body. It is worth noting that the RTP-Net achieves an average Dice of 0.93 ± 0.11 on 65 tasks with extensive samples. Specifically, 42 of 65 (64.6%) OARs segmentation tasks achieve satisfactory performance with a mean Dice of over 0.90, and 57 of 65 (87.7%) OARs segmentation tasks with a mean Dice of over 0.80. For OARs in the head (Fig. 3a), there are 20 of 27 (74.1%) OARs segmentation tasks achieving plausible performance with a mean Dice of over 0.80. For OARs in the chest (Fig. 3b), the lowest segmentation performance is found in the mediastinal lymph nodes with a mean Dice of 0.61, which may be due to their diffused and blurry boundaries. In addition, the Dice coefficients of segmentation results of all tested OARs in the abdomen (Fig. 3c) and pelvic cavity

**Table 1 | The detailed configuration for multi-resolution segmentation framework**

| Procedure | Design choice | Coarse model | Fine model |
|---|---|---|---|
| Pre-processing | Intensity normalization | If CT, z score with fixed mean and standard deviation (SD) & clipping to [−1, 1];<br>If MRI, percentile z score with mean and SD & clipping to [−1, 1] | If CT, z score with fixed mean and SD & clipping to [−1, 1];<br>If MRI, percentile z score with mean and SD & clipping to [−1, 1] |
| | Image resampling strategy | Nearest neighbor interpolation | Nearest neighbor interpolation;<br>Linear interpolation |
| | Annotation resampling strategy | [0, 1, ..., class-1] encoding nearest neighbor / linear interpolation | [0, 1, ..., class-1] encoding nearest neighbor interpolation |
| | Image target spacing | Spacing fixed to [5, 5, 5] | Spacing fixed to [1, 1, 1] |
| | Network topology | VB-Net for common organs;<br>Adaptive VB-Net for large organs | VB-Net for common organs;<br>Adaptive VB-Net for large organs |
| | Patch size | [96, 96, 96] | [96, 96, 96] for common organs;<br>[196, 196, 196] for large organs |
| | Batch size | At least 2, given multi-GPU memory constraint | At least 2, given multi-GPU memory constraint |
| Training | Learning rate | Step learning rate schedule (initial, 1e-4) | Step learning rate schedule (initial, 1e-4) |
| | Loss function | Dice and boundary Dice | Dice and boundary Dice for OARs;<br>3D Dice, boundary Dice, and adaptive 2D Dice for CTV and PTV |
| | Optimizer | Adam (momentum = 0.9, decay = 1e-4, betas = (0.9, 0.999)) | Adam (momentum = 0.9, decay = 1e-4, betas = (0.9, 0.999)) |
| | Data augmentation | Rotating, scaling, flipping, shifting & adding noise | Rotating, scaling, flipping, shifting & adding noise |
| | Training procedure | 1000 epochs, global sampling & mask sampling | 1000 epochs, global sampling & mask sampling |
| Testing | Configuration for pre-processing | Resampling to fix spacing as training;<br>Image partition given GPU memory | Available to expand the bounding box with user-set size or not;<br>Resampling to fix spacing as training;<br>Image partition given GPU memory |
| | Configuration for post-processing | Resampling to raw image spacing;<br>Available to pick the largest connected component (CC) in segmentation or not;<br>Available to remove small CC in segmentation or not | Resampling to raw image spacing;<br>Available to pick the largest CC in segmentation or not;<br>Available to remove small CC in segmentation or not |

(Fig. 3d) parts are higher than 0.80. Moreover, segmentations of the spinal cord, spinal canal, and external skin in the whole body also achieve superior agreement with manual ground truth. Note that the segmentation of external skin is assisted by the adaptive input module in the RTP-Net (Fig. 2b), due to its large size. In summary, the majority of the segmentation tasks achieve high accuracy by using the proposed RTP-Net, which verifies its superior segmentation performance. It should be outlined that auto-segmentation results will be reviewed and modified by the radiation oncologist to ensure accuracy and safety of RT.

To fully evaluate segmentation quality and efficiency of our proposed RTP-Net, three state-of-the-art methods, including U-Net, nnU-Net, and Swin UNETR, are included for comparison. Typical segmentation results of eight OARs (including brain, brainstem, rib, heart, liver, pelvis, rectum, and bladder) by four methods are provided in Fig. 4 for qualitative comparison. It can be seen that our RTP-Net achieves consistent segmentations with manual ground truth in all eight OARs, while the comparison methods show over- or under-segmentations. In particular, both U-Net and nnU-Net under-segment four OARs such as brainstem, rib, heart, and pelvis (Fig. 4a–d), while over-segment two OARs such as liver and bladder (Fig. 4e, f). For the remaining two OARs such as brain and rectum (Fig. 4g, h), U-Net and nnU-Net show different performances, with U-Net having under-segmentation while nnU-Net having over-segmentation. Swin UNETR achieves consistent segmentations with manual ground truth in the bladder and brain, while has under-segmentations in the other six OARs. It is worth emphasizing again that the inaccurate segmentation of OARs may influence subsequent steps of target tumor delineation and treatment planning, and finally the precise radiation therapy of the tumor. Overall, in comparison to U-Net, nnU-Net, and Swin UNETR, our proposed RTP-Net achieves comparable or superior results in segmenting OARs.

To quantitatively evaluate segmentation performance of RTP-Net, both Dice coefficient and average inference time are calculated. Figure 5a and Supplementary Table 2 show Dice coefficients on a set of segmentation tasks by four methods. It can be seen that the majority of segmentation tasks give high Dice coefficients, especially in segmentation of brain, liver, and pelvis with relatively less variation. Compared to nnU-Net, RTP-Net shows no significant difference in segmentation of most organs in terms of Dice coefficient, except rectum. While, compared to U-Net, RTP-Net shows significant difference in better segmenting brainstem, liver, and rectum. Besides, compared to Swin UNETR, RTP-Net shows better performance in segmentation of brainstem, heart, liver, and rectum. Overall, the average Dice coefficients of RTP-Net, U-Net, nnU-Net, and Swin UNETR in segmentation of eight OARs are $0.95 \pm 0.03$, $0.91 \pm 0.06$, $0.95 \pm 0.03$, and $0.94 \pm 0.03$, respectively. Results indicate that RTP-Net achieves comparable or more accurate segmentation performance than other methods, which is consistent with visual results given in Fig. 4.

In addition, the inference efficiency of four methods in the above eight OAR segmentation tasks is further evaluated in Fig. 5b, c and Supplementary Table 3. As a lightweight framework, RTP-Net takes less than 2 s in most segmentation tasks, while U-Net, nnU-Net, and Swin UNETR take 40–200 s, 200–2000 s, and 15–200 s, respectively. The heat map of inference time of four methods in segmentation tasks visually demonstrates a significant difference between RTP-Net and the other three methods. The ultra-high segmentation speed of RTP-Net can be attributed to the customized coarse-to-fine framework with multi-resolutions, which conducts coarse localization and fine segmentation sequentially and also reduces GPU memory cost significantly. In addition, the highly efficient segmentation capability of RTP-Net is also confirmed in more delineation experiments, as shown in Supplementary Fig. 2. Therefore, our proposed RTP-Net can achieve

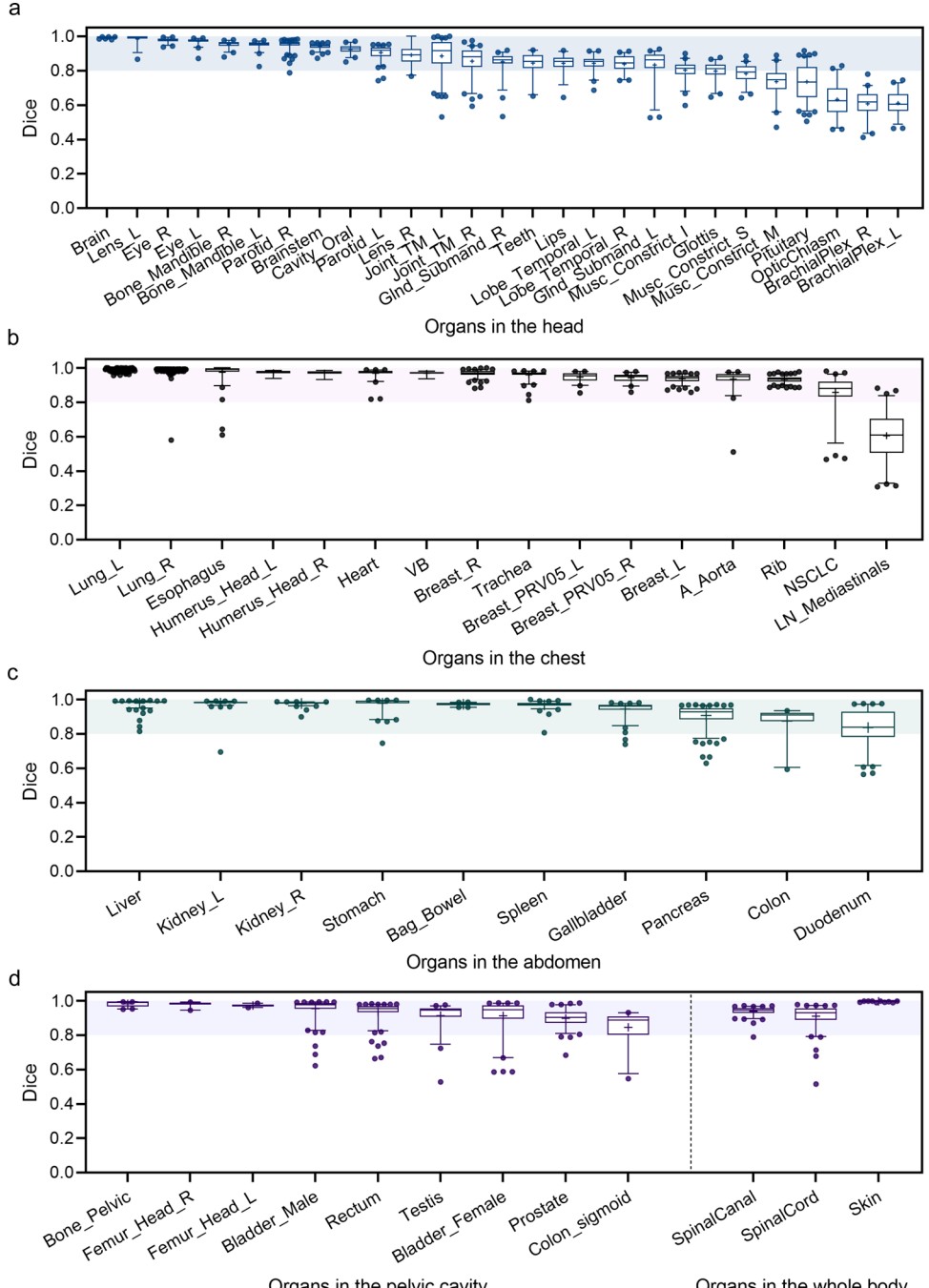

**Fig. 3 | The segmentation performance of the RTP-Net on whole-body OARs.**
The Dice coefficients in segmenting OARs in the head (**a**), chest (**b**), abdomen (**c**) parts, as well as those in the pelvic cavity part and whole body (**d**). The shadows in four box-and-whisker plots give the Dice coefficients with a range from 0.8 to 1.0. The first quartile forms the bottom and the third quartile forms the top of the box, in which the line and the plus sign represent the median and the mean values, respectively. The whiskers range from 2.5th to 97.5th percentile, and points below and above the whiskers are drawn as individual dots. The detailed number for each organ can be referred to Supplementary Fig. 1.

excellent segmentation performance, with superior accuracy and ultra-high inference speed.

**Segmentation of multiple OARs, CTV, and PTV by RTP-Net**
Given an input 3D image, we need to jointly segment all existing OARs (whether complete or partial), i.e., for delineation of the target volume, including CTV and PTV. Figure 6 illustrates segmentation results of multiple organs in each specific part, including head, chest, abdomen, and pelvic cavity. These results further verify performance of our RTP-Net.

Next, we evaluate performance of the target volume delineation model (Fig. 2c) to contour the target volumes, including CTV and PTV. In conventional clinical routine, PTV is generally obtained by dilating the CTV according to specific guidelines. Considering that the conventional dilated PTV are usually generated on specific software and may contain some errors (e.g., expanding beyond the skin or overlapping with OARs) that require manual corrections, an automatically generated PTV by RTP-Net can be quite convenient, save processing time, and show high precision with verified annotations from radiation oncologists. The delineation results of CTV and PTV for rectal cancer

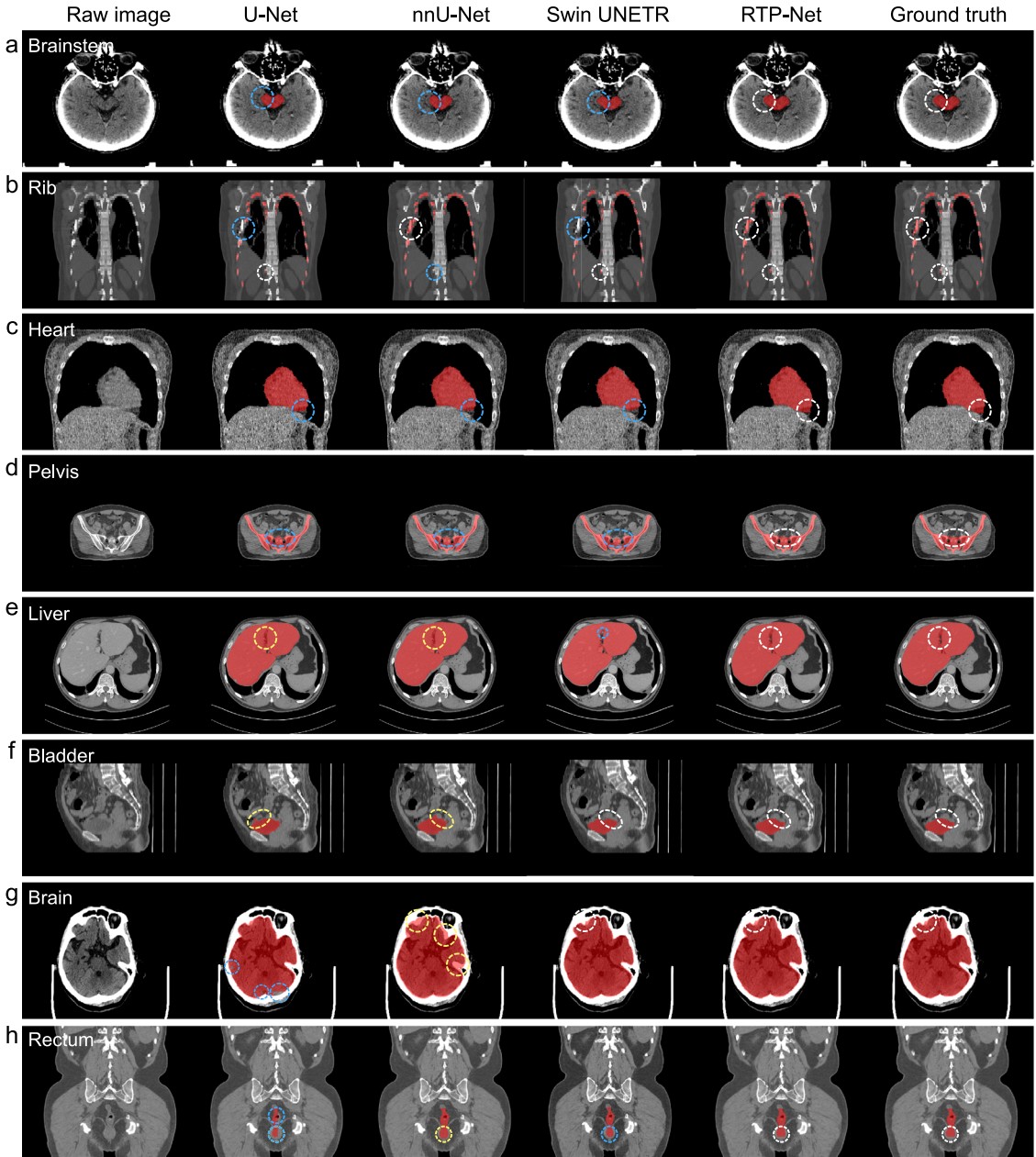

**Fig. 4 | Visual comparison of segmentation performance of our proposed RTP-Net, U-Net, nnU-Net, and Swin UNETR.** Segmentation is performed on eight OARs, i.e., (**a**) brainstem, (**b**) rib, (**c**) heart, (**d**) pelvis, (**e**) liver, (**f**) bladder, (**g**) brain, and (**h**) rectum. The white circles denote accurate segmentation compared to manual ground truth by four methods. The blue and yellow circles represent under-segmentation and over-segmentation, respectively.

are shown in Fig. 7 and Supplementary Table 4, using visual comparison, accuracy, and efficiency. As shown in Fig. 7a, the CTV delineation of the RTP-Net shows high performance compared with manual ground truth. Moreover, no significant difference in terms of Dice coefficient is found among the four segmentation methods (Fig. 7b). But, when comparing the mean inference time of CTV delineation, RTP-Net achieves the fastest delineation with less than 0.5 s ($0.40 \pm 0.05$ s), while U-Net, nnU-Net, and Swin UNETR take $108.41 \pm 19.38$ s, $248.43 \pm 70.38$ s, and $62.63 \pm 12.49$ s, respectively (Fig. 7c). A similar result is also found for the PTV delineation task, in which the inference times of RTP-Net, U-Net, nnU-Net, and Swin UNETR are $0.44 \pm 0.05$ s, $109.89 \pm 19.61$ s, $119.01 \pm 34.06$ s, and $92.65 \pm 16.03$ s, respectively. All these results (on CTV and PTV) confirm that the proposed RTP-Net can contour the target volume (including CTV and PTV) in a precise and fast manner. Segmentation results of OARs, as well as target tumor, can be seen in Fig. 7d, in which

the PTV of rectal cancer is delineated and surrounded by nearby OARs, such as bag bowel, pelvis, and vertebra. Note that, in our method, the boundary-aware attention map is adopted to avoid segmentation failure of the upper and lower boundaries of the target volume, by considering the surrounding OARs and their boundaries in our target volume delineation model. This could avoid the toxicity of radiation to normal organs, and makes the following dose simulation and treatment more precise.

So far, we have demonstrated that the proposed deep learning-based segmentation framework can automatically, efficiently and accurately delineate the OARs and target volumes. There are multiple AI-based software tools that are commercially available and have been used in clinical practices to standardize and accelerate the RT procedures. They include atlas-based contouring tool for automatic segmentation[12,34–37], and knowledge-based planning module for automatic treatment planning[38–40]. Here, we focus on exploring of AI-based

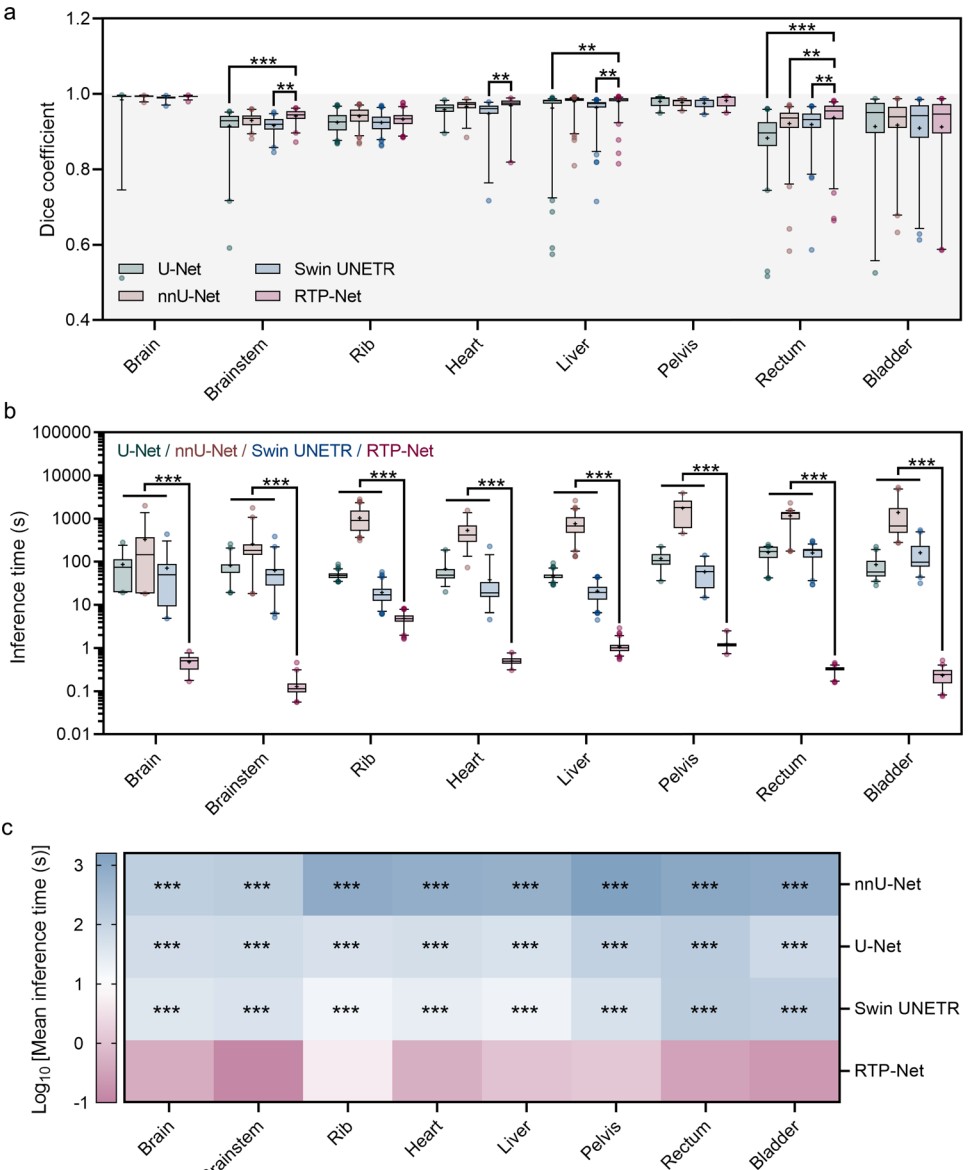

**Fig. 5 | Quantitative comparison of segmentation performance of four methods in terms of Dice coefficient and inference time. a** Dice coefficients of eight segmentation tasks by our proposed RTP-Net, U-Net, nnU-Net, and Swin UNETR. **b** Mean inference times in segmenting eight OARs by four methods. Both Dice coefficients (**a**) and inference times (**b**) are shown in box-and-whisker plots. The first quartile forms the bottom and the third quartile forms the top of the box, in which the line and the plus sign represent the median and the mean values, respectively. The whiskers range from 2.5th to 97.5th percentile, and points below and above the whiskers are drawn as individual dots. The number of eight organs can be referred to Supplementary Fig. 1. Statistical analyses in (**a**) and (**b**) are performed using two-way ANOVA followed by Dunnett's multiple comparisons tests. Asterisk represents two-tailed adjusted $p$ value, with * indicating $p < 0.05$, ** indicating $p < 0.01$, and *** indicating $p < 0.001$. The $p$ values of Dice coefficients in (**a**) between RTP-Net and other three methods (U-Net, nnU-Net, and Swin UNETR) are 0.596, 0.999, and 0.965 for brain segmentation, respectively; <0.001, 0.234, and 0.001 for brainstem segmentation, respectively; 0.206, 0.181, and 0.183 for rib segmentation, respectively; 0.367, 0.986, and 0.010 for heart segmentation, respectively; 0.002, 0.999, 0.003 for liver segmentation, respectively; 0.991, 0.900, and 0.803 for pelvic segmentation, respectively; <0.001, 0.010, and 0.003 for rectum segmentation, respectively; 0.999, 0.827, and 0.932 for bladder segmentation, respectively. All $p$ values in (**b**) between RTP-Net and other three methods in eight organs are lower than 0.001. **c** The heat map of the mean inference times in multiple segmentation tasks. Asterisk represents two-tailed adjusted $p$ value obtained in (**b**), with *** indicating $p < 0.001$, showing the statistical significance between RTP-Net and the other three methods.

automatic segmentation of target volumes and its integration into RT workflows. These AI solutions have reportedly achieved comparable performance with manual delineations in segmentation accuracy, with minor editing efforts needed[12,35]. However, majority of the studies were only evaluated on limited organs and data with specific acquisition protocols, which affects their clinical applicability when used in different hospitals or for different target volumes. Two studies have tried to address this challenge to improve the model generalizability[41,42]. Nikolov et al. applied 3D U-Net to delineate 21 OARs in head and neck

CT scans, and achieved expert-level performance[41]. The study was conducted on the training set (663 scans) and testing set (21 scans) from routine clinical practice, and validation set (39 scans) from two distinct open-source datasets. Oktay et al. incorporated the AI model into the existing RT workflow, and demonstrated that AI model could reduce contouring time while yielding clinical valid structural contours for both prostate and head-and-neck RT planning[42]. Their study involved 6 OARs for prostate cancer and 9 OARs for head-and-neck cancer, where experiments were conducted on a set of 519 pelvic and

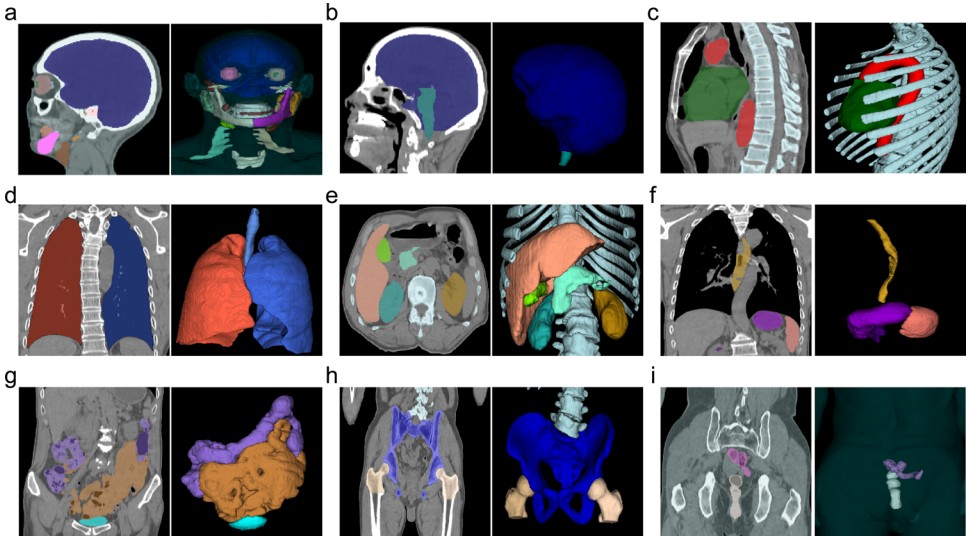

**Fig. 6 | Multiple organs-at-risk (OARs) segmentation results using the proposed RTP-Net. a** Brain, temporal lobe, eyes, teeth, parotid, mandible bone, larynx, brachial plexus; (**b**) brain, brainstem; (**c**) heart, trachea, rib, vertebra; (**d**) lungs; (**e**) liver, kidney, pancreas, gallbladder; (**f**) stomach, esophagus, spleen; (**g**) large bowel, small bowel, bladder; (**h**) femur head, bone pelvis; (**i**) testis, prostate. All samples are the CT images. In each sample, the left shows results in 2D view, and the right shows 3D rendering of segmented OARs.

242 head-and-neck CT scans acquired at eight distinct clinical sites with heterogeneous population groups and diverse image acquisition protocols. In contrast to previous works, we evaluate how RTP-Net can lead to generalized performance with extensive evaluation on 67 target volumes with varying volume sizes on a large-scale dataset of 28,581 cases (Supplementary Fig. 1). This large-scale dataset was obtained from eight distinct publicly-available datasets and one local dataset with varying acquisition settings and demographics (Supplementary Table 5). Our proposed model demonstrates performance generalizability across hospitals and target volumes, while achieving superior levels of agreement with expert contours and also time savings, which can facilitate easier deployment in clinical sites.

In addition, a variety of deep learning-based algorithms have been developed for automatically predicting the optimal dose distribution and accelerating the dose calculation[43,44]. It is speculated that integrating AI-assisted delineation and AI-aided dosimetric planning into the RTP system would largely promote the efficiency of RT and reduce workload in clinical practice, such as Pinnacle[3] (Philips Medical Systems, Madison, WI)[45]. The proposed RTP-Net was integrated into the CT-linac system (currently being tested for clinical use approval), supporting the All-in-One RT scheme, in which the auto-contouring results (reviewed by radiation oncologists) are used for dosimetric treatment planning, to maximize the dose delivered to the tumor while minimizing the dose to the surrounding OARs. This AI-accelerated All-in-One RT workflow has two potential merits: (1) AI-accelerated auto-contouring could remove systematic and subjective deviation, and ensure reproducible and precise decision, with the contouring time controlled within 15 s, much lower than the conventional contouring with 1–3 hour(s) or more, therefore, the total time for auto-contouring and manual editing by clinicians is much shorter than manual annotation from scratch; (2) All-in-One RT pipeline would be one-stop, incorporating multiple modules (i.e., auto-contouring) and making patients free of multiple turnaround waiting periods, and thus will greatly shorten the time of the whole process from days to minutes[32]. Importantly, multiple clinical steps in All-in-One RT workflow need human interventions and require the presence of dedicated staff (including radiation oncologist, dosimetrist, and medical physicist) to make decision, so there is an urgent need to improve the efficiency and save the turnaround time. In addition, in some clinical scenarios, there are more patients than what a hospital could accommodate, given that medical resources (e.g., RT equipment,

and professional staff) are relatively insufficient. In these cases, AI-accelerated All-in-One RT workflow holds great potential to reduce healthcare burden and benefit patients.

In conclusion, to overcome limitations of manual contouring in RTP system, such as long waiting time, low reproducibility, and low consistency, we have developed a deep learning-based framework (RTP-Net) for automatic contouring of the target tumor and OARs in a precise and efficient manner. First, we develop a coarse-to-fine framework to lower GPU memory and improve segmentation speed without reducing accuracy based on a large-scale dataset. Next, by redesigning the architecture, our proposed RTP-Net achieves high efficiency with comparable or superior segmentation performance on multiple OARs, compared to the state-of-the-art segmentation frameworks (i.e., U-Net, nnU-Net, Swin UNETR). Third, to accurately delineate the target volumes (CTV/PTV), the OAR-aware attention map, boundary-aware attention map, as well as multi-dimension loss function are combined into the training of the network to facilitate boundary segmentation. This proposed segmentation framework has been integrated into a CT-linac system and is currently being tested for clinical use approval[32]. And this AI-accelerated All-in-One RT workflow holds great potential in improving the efficiency, reproducibility, and overall quality of RT for patients with cancer.

## Methods
### Data
This study was approved by the Research Ethics Committee in Fudan University Shanghai Cancer Center, Shanghai, China (No. 2201250-16). A total of 362 images of rectal cancer were collected. Written informed consent was waived because of the retrospective nature of the study. The rest 28,219 data in experiments came from publicly available multi-center datasets (itemized in Supplementary Table 5), i.e., The Cancer Imaging Archive (TCIA, https://www.cancerimagingarchive.net/)[46], Head and Neck (HaN) Autosegmetation Challenge 2015 from Medical Image Computing and Computer Assisted Intervention society (MICCAI)[47,48], Segmentation of Thoracic Organs at Risk in CT Images (SegTHOR) Challenge 2019[49], Combined (CT-MR) Healthy Abdominal Organ Segmentation (CHAOS) Challenge 2019[50], Medical Segmentation Decathlon (MSD) Challenge from MICCAI 2018[51], and LUng Nodule Analysis (LUNA) 2016[52]. All the CT images were non-contrast-enhanced.

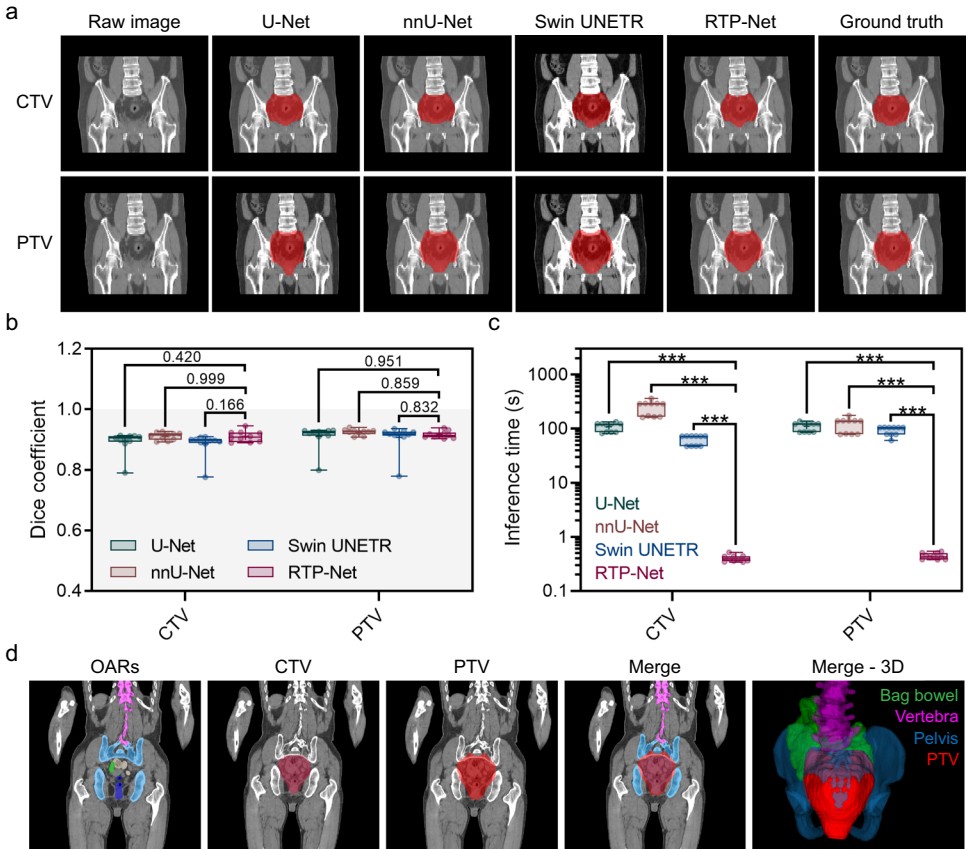

**Fig. 7 | The performance of target volume delineation by the proposed RTP-Net, compared with U-Net, nnU-Net, and Swin UNETR. a** Delineation results of the clinical target volume (CTV) and planning target volume (PTV) by the proposed RTP-Net, U-Net, nnU-Net, and Swin UNETR, labeled by red color. (**b**) Dice coefficients and (**c**) inference times of four methods in target volume delineation, shown in box-and-whisker plots. The first quartile forms the bottom and the third quartile forms the top of the box, in which the line and the plus sign represent the median and the mean values, respectively. The whiskers range from minimum to maximum showing all points. Statistical analyses in (**b**) and (**c**) are performed using two-way ANOVA followed by Dunnett's multiple comparison tests, with $n = 10$ replicates per condition. The two-tailed adjusted $p$ values of Dice coefficients in (**b**) between RTP-Net and other three methods (U-Net, nnU-Net, and Swin UNETR) are 0.420, 0.999, and 0.166 for CTV segmentation, respectively, while 0.951, 0.859, and 0.832 for PTV segmentation, respectively. All two-tailed adjusted $p$ values of inference times in (**c**) between RTP-Net and other three methods are lower than 0.001, indicated with ***. (**d**) Overview of the organs-at-risk (OARs) and target volumes. The segmentation results of PTV and neighboring bag bowel, vertebra, and pelvis are marked in red, green, pink, and blue, respectively.

**Data heterogeneity.** Supplementary Table 5 summarizes scanner types and acquisition protocols, with patient demographics provided in Supplementary Table 6. More details about datasets can be found in the corresponding references.

**Training and testing datasets.** In this study, we include a total of 28,581 cases for 67 segmentation tasks, covering whole-body organs and target tumors (Supplementary Fig. 1). In all the data, 23,728 cases are used as the training set (-83%), and the rest 4,853 cases are used as the testing set (-17%).

**Annotation protocols.** The ground truth of segmentation is obtained from manual delineations of experienced raters. The details are described as follows:

(1) Image data preparation. Large-scale images from multiple diverse datasets are adopted in this study (e.g., varying scanner types, populations, and medical centers) to lower the possible sampling bias. All CT images are in DICOM or NIFIT formats.

(2) Annotation tools. Based on raters' preferences, several widely used tools are adopted to annotate the target at pixel-level details and visualize them, i.e., ITK-SNAP 3.8.0 (http://www.itksnap.org/pmwiki/pmwiki.php) and 3D Slicer 5.0.2 (https://www.slicer.org/). These tools support both semi-automatic and manual annotation. Semi-automatic annotation can be used for annotation

initialization and followed by manual correction. This strategy can save the annotation efforts.

(3) Contouring protocol. For each annotation task, experienced raters and a senior radiation oncologist are involved. The corresponding consensus guidelines (e.g., RTOG guidelines) or anatomy textbooks are reviewed and a specific contouring protocol is made after discussion. Annotations are initially contoured by experienced raters and finally refined and approved by the senior radiation oncologist. Below we list the consensus guidelines.

**Head dataset.** A total of 27 anatomical structures are contoured. The anatomical definitions of 25 structures refer to the Brouwer atlas[53] and neuroanatomy textbook[54], i.e., brain, brainstem, eyes (left and right), parotid glands (left and right), bone mandibles (left and right), lens (left and right), oral cavity, joint TM (left and right), lips, teeth, submandibular gland (left and right), glottis, pharyngeal constrictor muscles (superior, middle, and inferior), pituitary, chiasm, and brachial plex (left and right). The contouring of temporal lobes (left and right) refers to the brain atlas[55].

**Chest dataset.** A total of 16 anatomical structures are contoured, in which 8 anatomical structures are defined following the Radiation Therapy Oncology Group (RTOG) guideline 1106[56] and the textbook of

cardiothoracic anatomy[57], i.e., heart, lungs (left and right), ascending aorta, esophagus, vertebral body, trachea, and rib. Breast (left and right), breast_PRV05 (left and right), mediastinal lymph nodes, and humerus head (left and right) are contoured referring to the RTOG breast cancer atlas[58]. Moreover, the contouring of NSCLC follows RTOG 0515[59].

**Abdomen dataset.** Ten anatomical structures are contoured (i.e., bowel bag, gallbladder, kidney (left and right), liver, spleen, stomach, pancreas, colon, and duodenum) referring to RTOG guideline[60], its official website for delineation recommendations (http://www.rtog.org), and Netter's atlas[61].

**Pelvic cavity dataset.** Nine anatomical structures are contoured referring to RTOG guideline[60] and Netter's atlas[61], including femur head (left and right), pelvis, bladder (male and female), rectum, testis, prostate, and colon_sigmoid.

**Whole body dataset.** The structures of the spinal canal, spinal cord, and external skin are also contoured referring to RTOG guideline 1106[56].

**Tumor dataset.** The contours of the CTV and PTV mainly refer to the RTOG atlas[62] and AGITG atlas[63].

## Image pre-processing
Considering the heterogeneous image characteristics from multiple centers, data pre-processing is a critical step to normalize data.

**Configuration of target spacing.** In the coarse-level model (low resolution), a large target spacing of $5 \times 5 \times 5$ mm$^3$ is recommended to obtain global location information, while, in the fine-level model (high resolution), we apply a small target spacing of $1 \times 1 \times 1$ mm$^3$ to acquire local structural information.

**Image resampling strategy.** In the training of the coarse-level model, the nearest-neighbor interpolation method is recommended to resample the image into the target spacing. In the training of the fine-level model, the nearest-neighbor interpolation and linear interpolation methods can be used for the resampling of anisotropic and isotropic images, respectively, to suppress the resampling artifacts.

**Configuration of patch size and batch size.** Patch size and batch size are usually limited by the given graphics processing unit (GPU) memory. For the segmentation of common organs, the patch size of $96 \times 96 \times 96$ is recommended for both the coarse-level model and the fine-level model. For segmentation of large organs, such as whole-body skin, the patch sizes of the coarse-level model and the fine-level model are $96 \times 96 \times 96$ and $196 \times 196 \times 196$, respectively. The mini-batch patches with fixed size are cropped from the resampled image by randomly generating center points in the image space.

**Intensity normalization.** Patches with target size and spacing could be normalized to the intensity of $[-1, 1]$, which can help the network converge quickly. For CT images, the intensity values are quantitative, which reflects physical property of tissue. Thus, fixed normalization is used, where each patch is normalized by subtracting the window level and then being divided by the half window width of the individual organ. After normalization, each patch is clipped to the range of $[-1, 1]$ and then fed to the network for training.

## Training settings
Our proposed framework allows setting individual learning rates and optimizer configurations based on specific tasks.

**Learning rate.** It is used to refine the network, where the learning rate could reduce from a large initial value to a small value with convergence of the network.

**Optimizer.** The Adam optimizer is used with adjustable hyper-parameters including momentum, decay, and betas.

**Data augmentation.** It is used to improve model robustness, including rotation, scaling, flipping, shifting, and adding noise.

**Training procedure.** To ensure robustness to class imbalance, two sampling schemes are adopted to generate mini-batches from one training image, including global sampling and mask sampling. Specifically, the global sampling scheme randomly generates center points in the entire foreground space, and the mask sampling scheme randomly generates center points in the regions of interest (ROIs). Global sampling is recommended for the coarse-level model to achieve the goal of locating the target ROI, and mask sampling is recommended for the fine-level model to achieve the goal of delineating the target volume accurately.

**Loss functions.** The basic segmentation loss functions, such as Dice, boundary Dice, and focal loss function, can be used to optimize the network. The multi-dimensional loss function is defined as the adaptive Dice loss function to enforce the network to pay attention to the boundary segmentation, especially the boundary of each 2D slice:

$$\text{loss}_{\text{adaptive}} = \lambda_1 \times \text{loss}_{3D} + \lambda_2 \times \sum_{i=1}^{n} \lambda_{\text{adaptive}}^{i} \times \text{loss}_{2D}^{i} \qquad (1)$$

In this equation, $\text{loss}_{3D}$ refers to 3D Dice loss and $\lambda_1$ is its weight, while $\text{loss}_{2D}^{i}$ refers to the 2D Dice loss of the i-th 2D slice and $\lambda_{\text{adaptive}}^{i}$ is its adaptive weight calculated from the performance of this 2D slice; $\lambda_2$ is the weight of 2D Dice loss. More detailed definitions of 3D Dice loss and 2D Dice loss are given in the following two equations:

$$\text{loss}_{3D} = 1 - \frac{2 \times \text{pred}_{3D} \times \text{target}_{3D}}{\text{pred}_{3D} + \text{target}_{3D}} \qquad (2)$$

$$\text{loss}_{2D}^{i} = 1 - \frac{2 \times \text{pred}_{2D}^{i} \times \text{target}_{2D}^{i}}{\text{pred}_{2D}^{i} + \text{target}_{2D}^{i}} \qquad (3)$$

In these two equations, $\text{pred}_{3D}$ denotes the 3D prediction and $\text{target}_{3D}$ denotes its manual ground truth, while $\text{pred}_{2D}^{i}$ denotes the 2D prediction of the i-th 2D slice and $\text{target}_{2D}^{i}$ denotes its manual ground truth. The settings of the hyper-parameters go as follows: $\lambda_1$ is set as 0.7, and $\lambda_2$ is set as 0.3. Besides, $\lambda_{\text{adaptive}}$ is an adaptive weight calculated from the following equation:

$$\lambda_{\text{adaptive}}^{i} = 1 - \left( \frac{2 \times \text{pred}_{2D}^{i} \times \text{target}_{2D}^{i}}{\text{pred}_{2D}^{i} + \text{target}_{2D}^{i}} \right)^2 \qquad (4)$$

Except for the multi-dimensional loss, the attention mechanisms (including the boundary-aware attention map and the OAR-aware attention map) are also specifically designed for the target volume delineation tasks. Detailed information is described in the Results and Discussion section.

## Network component: VB-Net
In our framework, VB-Net is a key component for multi-size organ segmentation. The VB-Net structure is composed of input block, down block, up block, and output block (Supplementary Fig. 3). The down/up blocks are implemented in form of residual structures, and the

bottleneck is adopted to reduce the dimension of feature maps. In each down/up block, the number of bottlenecks is available for the user to assign. Moreover, the skip connection is needed at each resolution level. Especially, VB-Net can also be customized to process large 3D image volumes, e.g., whole-body CT scans. In the customized VB-Net, an additional down-sampling operation before feeding the image to the backbone and an additional up-sampling operation after generating the segmentation probability maps are added to reduce GPU memory cost and enlarge the receptive field of the VB-Net at the same time. For these large organs with high-intensity homogeneity, the enlarged receptive field of the customized VB-Net contributes to focus on the boundaries with their surrounding low contrast organs.

### Inference configuration

The framework is implemented in PyTorch with one Nvidia Tesla V100 GPU. 10% of the training set is randomly selected as validation in each task, with its loss computed at the end of each training epoch. The training process is considered converged if the loss stops decreasing for 5 epochs. Also, the connected-component-based post-processing is supplied to eliminate spurious false positives by picking the largest connected component in the organ segmentation tasks or removing small connected components in the tumor segmentation tasks.

### Statistical analysis

For continuous variables that were approximately normally distributed, they were represented as mean ± standard deviation. For continuous variables with asymmetrical distributions, they were represented as median ($25^{th}$, $75^{th}$ percentiles). To quantitatively compare the segmentation performance (including Dice coefficients and inference times) of RTP-Net with other three methods (including U-Net, nnU-Net, and Swin UNETR), statistical analyses were performed using two-way ANOVA, followed by Dunnett's multiple comparison tests. Two-tailed adjusted $p$ values were obtained and represented with asterisk, with * indicating $p < 0.05$, ** indicating $p < 0.01$, and *** indicating $p < 0.001$. All statistical analyses were implemented using IBM SPSS 26.0.

Box-and-whisker plots were used to qualitatively compare the segmentation performance (including Dice coefficients and inference times) of RTP-Net with other three methods (including U-Net, nnU-Net, and Swin UNETR), which was plotted by GraphPad Prism 9. Visualization of segmentation results was generated with ITK-SNAP 3.8.0. All figures were created by Adobe Illustrator CC 2019.

### Reporting summary

Further information on research design is available in the Nature Research Reporting Summary linked to this article.

## Data availability

The OAR-related images (N = 28,219) that support experiments in this paper came from the publicly available multi-center datasets, i.e., The Cancer Imaging Archive (TCIA, https://www.cancerimagingarchive.net/), Head and Neck (HaN) Autosegmetation Challenge 2015 (https://paperswithcode.com/dataset/miccai-2015-head-and-neck-challenge), Segmentation of Thoracic Organs at Risk in CT Images (SegTHOR) Challenge 2019 (https://segthor.grand-challenge.org/), Combined (CT-MR) Healthy Abdominal Organ Segmentation (CHAOS) Challenge 2019 (https://chaos.grand-challenge.org/), Medical Segmentation Decathlon (MSD) Challenge 2018 (http://medicaldecathlon.com/), and LUng Nodule Analysis (LUNA) 2016 (https://luna16.grand-challenge.org/). The rest tumor-related data (N = 362) were obtained from Fudan University Shanghai Cancer Center (Shanghai, China), where partial data (i.e., 50 cases) are released together with the code, with the permission obtained from respective cancer center. The full dataset are protected because of privacy issues and regulation policies in cancer center.

## Code availability

The related code is available on GitHub (https://github.com/simonsf/RTP-Net)[64].

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

## Acknowledgements

The study is supported by the following funding: National Natural Science Foundation of China 62131015 (to Dinggang Shen) and 81830056 (to Feng Shi); Key R&D Program of Guangdong Province, China 2021B0101420006 (to Xiaohuan Cao, Dinggang Shen); Science and Technology Commission of Shanghai Municipality (STCSM) 21010502600 (to Dinggang Shen).

## Author contributions

Study conception and design: D.S., Y.G., and F.S.; Data collection and analysis: M.H., Q.Z., Y.W., Y.S., Y.C., Y.Y.; Interpretation of results: W.H., J.Wu, J.Wang, W.Z., J.Z., X.C., Y.Z., and X.S.Z.; Manuscript preparation: J.Wu, F.S., Q.Z., and D.S. All authors reviewed the results and approved the final version of the manuscript. F. Shi, W. Hu, and J. Wu contributed equally to this work.

## Competing interests

F.S., J.W., M.H., Q.Z., Y.W., Y.S., Y.C., Y.Y., X.C., Y.Z., X.S.Z., Y.G., and D.S. are employees of Shanghai United Imaging Intelligence Co., Ltd.; W.Z., and J.Z. are employees of Shanghai United Imaging Healthcare Co., Ltd. The companies have no role in designing and performing the surveillance and analyzing and interpreting the data. All other authors report no conflicts of interest relevant to this article.
