## [Peer Review File · Nature Communications]

Reviewers' Comments:

Reviewer #1:

Remarks to the Author:

The authors assessed novel segmentation algorithms for automatic delineation of OARs and target volumes. Overall, it is of interest, since the dataset used is extensive. However, shortcomings should be addressed:

1. The organs depicted in Figure 3 do in part not correspond to the legend.
2. PTVs are generated from CTVs depending on the institution's systematic and random setup errors. It does therefore not make sense to derive this automatically. Instead, the CTV ought to be segmented and the margin applied according to local standards.
3. The performance of the automatic segmentations should be evaluated by radiation oncologists, and additionally, the performance of re-segmentation should be addressed.
4. The performance of this framework compared to the numerous other frameworks around needs to be evaluated.

Reviewer #2:

Remarks to the Author:

Summary

The authors propose a CNN based CT image segmentation model for radiotherapy planning, which automatically delineates organ boundaries surrounding target tumors. The proposed model is trained and evaluated on a large-scale dataset (28k) including different body parts.

The authors evaluated their solution in a very thorough manner displaying strong segmentation performance with their approach. In that regard, it's worth mentioning the engineering effort the authors have put in. In below, I would like to highlight some areas of improvements in case the authors would like to revise the manuscript and resubmit it in the future.

Evaluation setup

- The authors do not describe the heterogeneity of the dataset, e.g., whether it's coming from multiple centers, different scanner types (contrast, no-contrast), ethnic and age groups. Without such details and grouped analysis, it's difficult to reach any conclusion that the proposed solution generalize to images acquired from diverse set of population and can be safely applied in clinical practice. Especially compared to previous literature [1, 2], this is a big handicap. There are public benchmarks available which could provide more insights [1].

- Similarly, it would be good to include the annotation protocol utilized to manually annotate these images at scale. For instance, does it conform with the standard annotation protocols?

- I am afraid the results displayed in Figure 3, 4, and 5 are not easy to interpret due to the limited resolution of the images. Scientific rigor would be easier to assess if they were presented in tables with quantitative scores including confidence bounds obtained with different random seeds.

- The authors aggregate the results across all the OARs and report a single dice score. I think a distribution across organs would scientifically be more valuable. More importantly, the authors claim that 80% dice score is a plausible performance. Is this information validated by a radiation oncologist, and does it mean that we do not require any manual edits on the auto-generated contours for treatment planning?

Methodology

- There is no clear evidence or motivation explaining why the proposed approach should actually yield a better performance than U-Net or nn-Unet. It would be better to provide a clearer verbal explanation supported by higher quality visual explanations.

- If the memory or patch size are the main motivations for a two-stage approach, the prior art has already shown that full volume training and testing can be achieved by model and data parallelization at scale by using a large number of GPUs. Perhaps it might be good to use the publicly available GitHub repositories to test this.

- Line 66 – “accurate and robust delineation”. The paper does not talk about robustness as in resilience to imaging artefacts or population groups etc. It is probably better to rephrase it as “accurate and consistent delineation” or “reproducible”

Clinical utility

Although the computation time of model is an important aspect, manual editing time or revision is equally important as the model run-time. Clinicians shouldn't be expected to spend more time correcting the model errors than annotating them from scratch. In that regard, the manuscript is not providing an extensive answer to the question whether the auto-generated contours help clinicians actually save time. The prior art has already demonstrated the clinical utility in this regard on multi-center datasets in addition to the technical advancements.

[1] “Clinically Applicable Segmentation of Head and Neck Anatomy for Radiotherapy: Deep Learning Algorithm Development and Validation Study”, JMIR 2021.

[2] “Evaluation of deep learning to augment image-guided radiotherapy for head and neck and prostate cancers”. JAMA Network Open, 2020

REVIEWER COMMENTS

Reviewer #1 (Remarks to the Author):

The authors assessed novel segmentation algorithms for automatic delineation of OARs and target volumes. Overall, it is of interest, since the dataset used is extensive. However, shortcomings should be addressed:

Response:

We thank this reviewer for the valuable comments. We have revised the manuscript thoroughly according to the suggestions. The corresponding responses are listed as follows.

1. The organs depicted in Figure 3 do in part not correspond to the legend.

Response:

Thank you for pointing this out. Considering the limited space, we previously showed only the segmentation results for a few cases. To reduce the possible ambiguity, we removed all the organs in Figure 3 to focus on the quantitative metrics. We refer the readers to check **Figures 4 and 6**, where segmentation results are better demonstrated.

2. PTVs are generated from CTVs depending on the institution's systematic and random setup errors. It does therefore not make sense to derive this automatically. Instead, the CTV ought to be segmented and the margin applied according to local standards.

Response:

Thank you for pointing this out. We discussed this issue with our clinical partners. There are several reasons that automatically generated PTV is still useful. First, in conventional clinical routines, PTV is generally obtained by dilating the CTV according to specific guidelines. Considering there are still efforts that perform this dilating by using specific software, an automatically generated PTV could be quite convenient and save processing time. Second, the conventional dilated PTV usually contains some errors, such as expanding beyond the skin or overlapping with OARs, which require manual corrections. In contrast, the automatically generated PTV may not have this problem as the AI segmentation results show high precision with verified annotations from radiation oncologists. Third, the conventional PTV generation could serve as a backup plan, and it could be used in some corner cases where

AI results are not satisfactory. In conclusion, we explored providing the automated generated PTV as an option, and radiation oncologists could choose it or use the conventional PTV.

3. The performance of the automatic segmentations should be evaluated by radiation oncologists, and additionally, the performance of re-segmentation should be addressed.

Response:

Many thanks for your valuable suggestions. We are sorry for the unclear description. For the dataset, we first got the images and the corresponding ground truth annotations from radiation oncologists. Then the dataset was split into the training set and testing test. The AI model was constructed by using the training set and then evaluated on the testing set. Thus the performance we reported is the comparison between AI and radiation oncologists.

In practice, we adopted a human-in-the-loop (HITL) strategy to reduce the efforts of radiation oncologists and iteratively refine the deep learning model. At first, a small number of images were manually delineated by radiation oncologists to train the model. Then, the initial model was applied to new data and the generated segmentation results were corrected by radiation oncologists. The newly corrected images were added to the database and fed into the model for refinement. After 2-3 iterations, this deep learning model could achieve satisfactory segmentation performance. Details of the HITL strategy can be referred to our previous work [1-2]. In the submitted manuscript, we presented the final segmentation results of the model after several iterations.

[1] Wang, Y. et al. Quantitative analysis of chest CT imaging findings with the risk of ARDS in COVID-19 patients: A preliminary study. *Ann. Transl. Med.* **8**, 594 (2020). doi: 10.21037/atm-20-3554

[2] Shan, F. et al. Abnormal lung quantification in chest CT images of COVID-19 patients with deep learning and its application to severity prediction. *Med. Phys.* **48**, 1633-1645 (2021). doi: 10.1002/mp.14609

4. The performance of this framework compared to the numerous other frameworks around needs to be evaluated.

Response:

Thank you for the suggestion. In the previous manuscript, we compared the performance of our proposed RTP-Net with U-Net and nnU-Net [3]. Here, U-Net is the standard network backbone widely used in the image segmentation field. The nnU-Net is an established framework that supports out-of-the-box use and achieves state-of-the-art segmentation performance in many challenges. For example, it ranked 1st place in the Medical Segmentation Decathlon [4] in MICCAI'18 where 10 organs were to be segmented including the brain, lung, liver, pancreas, prostate, etc.

In the revised manuscript, we introduced another framework, named Swin UNETR [5], as an additional comparison method. It was chosen as recently Transformer methods are gradually used in the medical image analysis field as novel techniques, and Swin UNETR is the combination of Swin Transformer and U-Net. We updated **Figures 4, 5, and 7** (attached below). Results showed that the performance of the proposed method was comparable to Swin UNETR, while requiring a significantly shorter processing time.

[3] Isensee, F. et al. nnU-Net: A self-configuring method for deep learning-based biomedical image segmentation. *Nat. Methods* **18**, 203-211 (2021). doi:10.1038/s41592-020-01008-z

[4] Antonelli, M. et al. The Medical Segmentation Decathlon. *Nat. Commun.* **13**, 4128 (2022). doi:10.1038/s41467-022-30695-9

[5] Hatamizadeh, A. et al. Swin UNETR: Swin Transformers for Semantic Segmentation of Brain Tumors in MRI Images. *arXiv preprint arXiv: 2201.01266* (2022). doi:10.1007/978-3-031-08999-2_22

Figure 4. Visual comparison of segmentation performance of our proposed RTP-Net, U-Net, nnU-Net, and Swin UNETR. Segmentation is performed on eight OARs, i.e., (a) brainstem, (b) rib, (c) heart, (d) pelvis, (e) liver, (f) bladder, (g) brain, and (h) rectum. The white circles denote accurate segmentation compared to manual ground truth by four methods. The blue and yellow circles represent under-segmentation and over-segmentation, respectively.

Figure 5. Quantitative comparison of segmentation performance of four methods in terms of Dice coefficient and inference time. (a) Dice coefficients of eight segmentation tasks by our proposed RTP-Net, U-Net, nnU-Net, and Swin UNETR. Statistical analyses are performed using two-way ANOVA, with *** indicating $p < 0.001$, ** indicating $p < 0.01$, and * indicating $p < 0.05$. (b) Mean inference times in segmenting eight OARs by four methods. The error bar represents the 95% confidence interval (CI). (c) The heat map of the mean inference times in multiple tasks. Statistical analyses are performed between RTP-Net and the other three methods, using the t-test, with *** indicating $p < 0.001$.

Figure 7. The performance of target volume delineation by the proposed RTP-Net, compared with U-Net, nnU-Net, and Swin UNETR. (a) Visual comparison of target volume delineation results by the proposed RTP-Net, U-Net, nnU-Net, and Swin UNETR. (b) Dice coefficient and (c) inference time of four methods in target volume delineation. Statistical analyses are performed using two-way ANOVA, with *** indicating $p < 0.001$, ** indicating $p < 0.01$, and * indicating $p < 0.05$. (d) Overview of the OARs and target volumes.

Reviewer #2:

The authors propose a CNN based CT image segmentation model for radiotherapy planning, which automatically delineates organ boundaries surrounding target tumors. The proposed model is trained and evaluated on a large-scale dataset (28k) including different body parts. The authors evaluated their solution in a very thorough manner displaying strong segmentation performance with their approach. In that regard, it's worth mentioning the engineering effort the authors have put in. In below, I would like to highlight some areas of improvements in case the authors would like to revise the manuscript and resubmit it in the future.

Response:

Many thanks for the reviewer's affirmation. We have enclosed point-by-point responses to the reviewer's comments and revised the manuscript.

Evaluation setup

- The authors do not describe the heterogeneity of the dataset, e.g., whether it's coming from multiple centers, different scanner types (contrast, no-contrast), ethnic and age groups. Without such details and grouped analysis, it's difficult to reach any conclusion that the proposed solution generalize to images acquired from diverse set of population and can be safely applied in clinical practice. Especially compared to previous literature [1, 2], this is a big handicap. There are public benchmarks available which could provide more insights [1].

[1] "Clinically Applicable Segmentation of Head and Neck Anatomy for Radiotherapy: Deep Learning Algorithm Development and Validation Study", JMIR 2021.

[2] "Evaluation of deep learning to augment image-guided radiotherapy for head and neck and prostate cancers". JAMA Network Open, 2020.

Response:

Thank you for the valuable suggestion, which is closely related to the generalization of the model. We have added **Tables S5 and S6** for the dataset heterogeneity information. Briefly, most images came from the publicly available multi-center datasets (itemized in **Table S5**), and the images of rectal cancer were provided by a local hospital (Fudan University Shanghai Cancer Center, Shanghai, China). All the CT images were non-contrast-enhanced.

Table S5 summarized scanner types and acquisition protocols, with patient demographics provided in **Table S6**. More details about datasets could be found in the corresponding references. In conclusion, the use of diverse scanners and acquisition protocols could largely promote the generalization of the model.

All these descriptions and references have been added to the **Methods** and **Supplementary Information**.

Table S5. Imaging datasets used in this study.

Dataset	Acquisition site	Region	Scanner type	Organ part
TCIA ¹				
• Head-Neck-PET-CT	Hôpital général juif (HGJ) de Montréal, QC, Canada; Centre hospitalier universitaire de Sherbrooke (CHUS), QC, Canada; Hôpital Maisonneuve-Rosemont (HMR) de Montréal, QC, Canada; Centre hospitalier de l'Université de Montréal (CHUM), QC, Canada	North America	GE	Head
• NSCLC Radiogenomics ¹⁹	Stanford University Medical Center; Palo Alto Veterans Affairs Healthcare System	North America	GE	Chest
• A whole-body CT dataset	University Hospital Tübingen, Germany	Europe	Siemens Biograph mCT	ALL
HaN (MICCAI 2015) ^{2,3}	Harvard Medical School, Massachusetts General Hospital, USA	North America	Anonymous	Head
SegTHOR 2019 ⁴	Henri Becquerel Center (CHB), Rouen, France	Europe	Anonymous	Chest
CHAOS 2019 ⁵	Dokuz Eylul University Hospital, Izmir, Turkey	Europe	Philips SecuraCT; Philips Mx8000 CT; Toshiba; AquilionOne	Chest; Abdomen

MSD (MICCAI 2018) ⁶	Ludwig Maximilian University of Munich, Germany; Radboud University Medical Center of Nijmegen, The Netherlands; Polytechnique and CHUM Research Center Montreal, Canada; Tel Aviv University, Israel; Sheba Medical Center, Israel; IRCAD Institute Strasbourg, France; Hebrew University of Jerusalem, Israel; Memorial Sloan Kettering Cancer Center, USA	Europe; North America; Asia	GE	Chest; Abdomen
LUNA16 ⁷	Weill Cornell Medical College, USA; University of California, Los Angeles, USA; University of Chicago, USA; University of Iowa, USA; University of Michigan, USA	North America	GE; Philips; Siemens; Toshiba	Chest (NSCLC)
Local dataset	Fudan University Shanghai Cancer Center	Asia	uRT-linac 506c	Tumor volume (CTV and PTV)

Table S6. Patient demographics of the imaging datasets.

Dataset	Patient age (years)	Patient sex
TCIA ¹		
• Head-Neck-PET-CT	63 (18 ~ 90)	Male 76%; Female 24%
• NSCLC Radiogenomics ¹⁹	68 (24 ~ 87)	Male 64%; Female 36%
• A whole-body CT dataset	59 (11 ~ 95)	Male 44%; Female 56%
HaN (MICCAI 2015) ^{2,3}	57 (31 ~ 79)	Male 88%; Female 12%
SegTHOR 2019 ⁴	Anonymous	Anonymous
CHAOS 2019 ⁵	45 (18 ~ 63)	Male 55%; Female 45%
MSD ⁶	Anonymous	Anonymous
LUNA16 ⁷	59 (14 ~ 85)	Male 51%; Female 49%
Local dataset	Anonymous	Anonymous

- Similarly, it would be good to include the annotation protocol utilized to manually annotate these images at scale. For instance, does it conform with the standard annotation protocols?

Response:

We thank the reviewer's valuable suggestion. The details of the manual annotation process have been added to the **Methods**.

Extracted texts from the revised manuscript:

“(1) Image data preparation. Large-scale images from multiple diverse datasets are adopted in this study (e.g., varying scanner types, populations, and medical centers) to lower the possible sampling bias. All CT images are in DICOM or NIFIT formats.

(2) Annotation tools. Based on raters' preferences, several widely used tools are adopted to annotate the target at pixel-level details, i.e., ITK-SNAP (<http://www.itksnap.org/pmwiki/pmwiki.php>), 3D Slicer (<https://www.slicer.org/>), and MIMICS (Materialize, Leuven, Belgium). These tools support both semi-automatic and manual annotation. Semi-automatic annotation can be used for annotation initialization and followed by manual correction. This strategy can save the annotation efforts.

(3) *Contouring protocol.* For each annotation task, experienced raters and a senior radiation oncologist are involved. The corresponding consensus guidelines (e.g., RTOG guidelines) or anatomy textbooks are reviewed and a specific contouring protocol is made after discussion. Annotations are initially contoured by experienced raters and finally refined and approved by the senior radiation oncologist. Below we list the consensus guidelines.

Head Dataset. A total of 27 anatomical structures are contoured. The anatomical definitions of 25 structures refer to the Brouwer atlas⁸ and neuroanatomy textbook⁹, i.e., brain, brainstem, eyes (left and right), parotid glands (left and right), bone mandibles (left and right), lens (left and right), oral cavity, joint TM (left and right), lips, teeth, submandibular gland (left and right), glottis, pharyngeal constrictor muscles (superior, middle, and inferior), pituitary, chiasm, and brachial plex (left and right). The contouring of temporal lobes (left and right) refers to the brain atlas¹⁰.

Chest Dataset. A total of 16 anatomical structures are contoured, in which 8 anatomical structures are defined following the Radiation Therapy Oncology Group (RTOG) guideline 1106¹¹ and the textbook of cardiothoracic anatomy¹², i.e., heart, lungs (left and right), ascending aorta, esophagus, vertebral body, trachea, and rib. Breast (left and right), breast_PRV05 (left and right), mediastinal lymph nodes, and humerus head (left and right) are contoured referring to the RTOG breast cancer atlas¹³. Moreover, the contouring of NSCLC follows RTOG 0515¹⁴.

Abdomen Dataset. Ten anatomical structures are contoured (i.e., bowel bag, gallbladder, kidney (left and right), liver, spleen, stomach, pancreas, colon, and duodenum) referring to RTOG guideline¹⁵, its official website for delineation recommendations (<http://www.rtog.org>), and Netter's atlas¹⁶.

Pelvic Dataset. Nine anatomical structures are contoured referring to RTOG guideline¹⁵ and Netter's atlas¹⁶, including femur head (left and right), pelvis, bladder (male and female), rectum, testis, prostate, and colon_sigmoid.

Whole Body Dataset. The structures of the spinal canal, spinal cord, and external skin are also contoured referring to RTOG guideline 1106¹¹.

Tumor Dataset. The contours of the CTV and PTV mainly refer to the RTOG atlas¹⁷ and AGITG atlas¹⁸. ”

- I am afraid the results displayed in Figure 3, 4, and 5 are not easy to interpret due to the limited resolution of the images. Scientific rigor would be easier to assess if they were presented in tables with quantitative scores including confidence bounds obtained with different random seeds.

Response:

Many thanks to the reviewer for this valuable suggestion. First, we have increased the resolution of all images shown in **Figure 3-5** from 300 dpi to 1200 dpi. Second, as suggested, we have added the corresponding tables to quantitatively show the segmentation performance (i.e., Dice coefficients and inference times) of four methods. The results are shown below and added to the **Supplementary Information** of the revised manuscript.

Table S2. Dice coefficients of eight segmentation tasks by our proposed RTP-Net, U-Net, nnU-Net, and Swin UNETR. The dice coefficient is represented with mean and 95% CI.

	RTP-Net	U-Net	nnU-Net	Swin UNETR	$P_{(RTP-Net vs. U-Net)}$	$P_{(RTP-Net vs. nnU-Net)}$	$P_{(RTP-Net vs. Swin UNETR)}$
Brain	0.993 (0.992, 0.994)	0.901 (0.847, 0.956)	0.994 (0.993, 0.995)	0.976 (0.946, 1.000)	0.596	0.999	0.965
Brainstem	0.941 (0.938, 0.945)	0.915 (0.903, 0.926)	0.930 (0.926, 0.934)	0.916 (0.912, 0.921)	< 0.001	0.234	0.001
Rib	0.939 (0.936, 0.941)	0.925 (0.921, 0.928)	0.941 (0.938, 0.945)	0.924 (0.921, 0.928)	0.206	0.181	0.183
Heart	0.969 (0.962, 0.976)	0.928 (0.893, 0.963)	0.967 (0.962, 0.971)	0.947 (0.937, 0.958)	0.367	0.986	0.010
Liver	0.980 (0.977, 0.983)	0.963 (0.953, 0.973)	0.980 (0.976, 0.983)	0.964 (0.959, 0.969)	0.002	0.999	0.003
Pelvis	0.982 (0.978, 0.987)	0.980 (0.976, 0.984)	0.977 (0.955, 0.987)	0.976 (0.972, 0.979)	0.991	0.900	0.803
Rectum	0.937 (0.928, 0.946)	0.824 (0.795, 0.853)	0.921 (0.913, 0.930)	0.906 (0.887, 0.926)	< 0.001	0.010	0.003
Bladder	0.892 (0.861, 0.923)	0.804 (0.750, 0.859)	0.903 (0.877, 0.928)	0.889 (0.856, 0.923)	0.999	0.827	0.932

Table S3. Inference times (in second) in segmenting eight OARs by our proposed RTP-Net, U-Net, nnU-Net, and Swin UNETR. Time is represented with mean and 95% CI.

	RTP-Net	U-Net	nnU-Net	Swin UNETR	$P_{(RTP-Net vs. U-Net)}$	$P_{(RTP-Net vs. nnU-Net)}$	$P_{(RTP-Net vs. Swin UNETR)}$
Brain	0.48 (0.44, 0.52)	86.27 (69.97, 102.57)	328.30 (224.54, 432.06)	70.84 (50.33, 91.36)	< 0.001	< 0.001	< 0.001
Brainstem	0.13 (0.11, 0.14)	81.58 (71.03, 92.13)	256.93 (196.06, 317.80)	62.60 (50.41, 74.79)	< 0.001	< 0.001	< 0.001
Rib	4.87 (4.65, 5.09)	48.10 (46.69, 49.52)	1033.82 (947.71, 1119.93)	19.24 (17.77, 20.71)	< 0.001	< 0.001	< 0.001
Heart	0.51 (0.48, 0.53)	68.22 (56.66, 79.78)	1573.83 (1036.13, 2111.54)	38.28 (28.60, 47.96)	< 0.001	< 0.001	< 0.001
Liver	1.08 (1.03, 1.13)	46.70 (45.23, 48.17)	761.61 (699.95, 823.28)	20.79 (19.29, 22.30)	< 0.001	< 0.001	< 0.001
Pelvis	1.28 (1.16, 1.39)	119.24 (105.85, 132.63)	1845.30 (1486.18, 2204.41)	57.88 (48.71, 67.06)	< 0.001	< 0.001	< 0.001
Rectum	0.32 (0.31, 0.33)	164.76 (155.90, 173.62)	1163.37 (1102.64, 1224.10)	159.98 (151.43, 168.52)	< 0.001	< 0.001	< 0.001
Bladder	0.23 (0.21, 0.25)	85.31 (74.36, 96.26)	1379.01 (1083.79, 1674.23)	161.54 (135.26, 187.82)	< 0.001	< 0.001	< 0.001

Table S4. Dice coefficients and inference times (in second) of four methods in target volume delineation. All descriptions are represented with mean and 95% CI.

		RTP-Net	U-Net	nnU-Net	Swin UNETR	$P_{(RTP-Net vs. U-Net)}$	$P_{(RTP-Net vs. nnU-Net)}$	$P_{(RTP-Net vs. Swin UNETR)}$
Dice coefficient	CTV	0.910 (0.897, 0.923)	0.893 (0.866, 0.919)	0.911 (0.902, 0.920)	0.885 (0.857, 0.913)	0.420	0.999	0.166
	PTV	0.916 (0.908, 0.924)	0.910 (0.882, 0.939)	0.925 (0.918, 0.932)	0.907 (0.874, 0.939)	0.951	0.859	0.832
Inference time (s)	CTV	0.40 (0.36, 0.44)	108.41 (93.80, 123.02)	248.43 (195.36, 301.50)	62.63 (53.21, 72.05)	< 0.001	< 0.001	< 0.001
	PTV	0.44 (0.40, 0.48)	109.89 (95.10, 124.68)	119.01 (93.33, 144.70)	92.65 (80.56, 104.74)	< 0.001	< 0.001	< 0.001

- The authors aggregate the results across all the OARs and report a single dice score. I think a distribution across organs would scientifically be more valuable. More importantly, the authors claim that 80% dice score is a plausible performance. Is this information validated by a radiation oncologist, and does it mean that we do not require any manual edits on the auto-generated contours for treatment planning?

Response:

Thanks for the reviewer’s suggestion. We have added detailed performance information in **Table S1**. It can be easily found that 42 of 65 (64.6%) OARs segmentation tasks achieve satisfactory performance with a mean Dice of over 0.90, and 57 of 65 (87.7%) OARs segmentation tasks with a mean Dice of over 0.80. Moreover, we need to note that the auto-contouring step would be followed by the radiation oncologist’s review (with minimal required modification) for treatment planning. We have emphasized this detailed information in the main text.

Table S1. The Dice coefficients of RTP-Net in segmenting whole-body OARs. Each Dice coefficient is represented with a mean and 95% confidence interval (CI).

No.	Head part	Dice coefficient	No.	Chest part	Dice coefficient
1	Brain	0.993 (0.992, 0.994)	1	Lung_L	0.988 (0.988, 0.989)
2	Lens_L	0.985 (0.975, 0.995)	2	Lung_R	0.988 (0.988, 0.989)
3	Eye_R	0.977 (0.974, 0.981)	3	Esophagus	0.975 (0.962, 0.988)
4	Eye_L	0.972 (0.966, 0.977)	4	Humerus_Head_L	0.972 (0.961, 0.983)
5	Bone_Mandible_R	0.952 (0.946, 0.958)	5	Humerus_Head_R	0.971 (0.960, 0.982)
6	Bone_Mandible_L	0.951 (0.944, 0.959)	6	Heart	0.969 (0.962, 0.976)
7	Parotid_R	0.951 (0.947, 0.956)	7	VB	0.969 (0.964, 0.975)
8	Brainstem	0.941 (0.938, 0.945)	8	Breast_R	0.968 (0.964, 0.971)
9	Cavity_Oral	0.924 (0.916, 0.932)	9	Trachea	0.960 (0.954, 0.965)
10	Parotid_L	0.905 (0.897, 0.914)	10	Breast_PRV05_L	0.947 (0.939, 0.954)
11	Lens_R	0.892 (0.871, 0.912)	11	Breast_PRV05_R	0.942 (0.935, 0.950)
12	Joint_TM_L	0.886 (0.866, 0.906)	12	Breast_L	0.937 (0.933, 0.941)
13	Joint_TM_R	0.856 (0.838, 0.874)	13	A_Aorta	0.934 (0.914, 0.954)
14	GlnD_Submand_R	0.852 (0.833, 0.872)	14	Rib	0.933 (0.930, 0.935)

15	Teeth	0.845 (0.816, 0.873)	15	NSCLC	0.858 (0.834, 0.883)
16	Lips	0.844 (0.827, 0.861)	16	LN_Mediastinals	0.606 (0.571, 0.640)
17	Lobe_Temporal_L	0.843 (0.830, 0.857)			
18	Lobe_Temporal_R	0.840 (0.826, 0.853)	No.	Abdomen part	Dice coefficient
19	GlnD_Submand_L	0.833 (0.805, 0.861)	1	Liver	0.980 (0.977, 0.983)
20	Musc_Constrict_I	0.804 (0.788, 0.821)	2	Kidney_L	0.979 (0.972, 0.985)
21	Glottis	0.798 (0.782, 0.815)	3	Kidney_R	0.978 (0.975, 0.980)
22	Musc_Constrict_S	0.784 (0.768, 0.800)	4	Stomach	0.978 (0.971, 0.985)
23	Musc_Constrict_M	0.737 (0.710, 0.763)	5	Bag_Bowel	0.973 (0.970, 0.976)
24	Pituitary	0.736 (0.717, 0.754)	6	Spleen	0.969 (0.965, 0.973)
25	OpticChiasm	0.632 (0.602, 0.662)	7	Gallbladder	0.944 (0.936, 0.953)
26	BrachialPlex_R	0.607 (0.586, 0.629)	8	Pancreas	0.907 (0.898, 0.916)
27	BrachialPlex_L	0.603 (0.578, 0.628)	9	Colon	0.874 (0.839, 0.910)
			10	Duodenum	0.837 (0.818, 0.857)
No.	Pelvic cavity part	Dice coefficient	No.	Whole body	Dice coefficient
1	Bone_Pelvic	0.982 (0.978, 0.987)	1	SpinalCanal	0.939 (0.934, 0.944)
2	Femur_Head_R	0.981 (0.978, 0.985)	2	SpinalCord	0.911 (0.897, 0.924)
3	Femur_Head_L	0.973 (0.970, 0.975)	3	External_Skin	0.997 (0.997, 0.997)
4	Bladder_Male	0.955 (0.944, 0.966)			
5	Rectum	0.937 (0.928, 0.946)			
6	Testis	0.913 (0.890, 0.937)			
7	Bladder_Female	0.902 (0.874, 0.931)			
8	Prostate	0.899 (0.888, 0.909)			
9	Colon_sigmoid	0.846 (0.805, 0.886)			

Methodology

- There is no clear evidence or motivation explaining why the proposed approach should actually yield a better performance than U-Net or nnU-Net. It would be better to provide a clearer verbal explanation supported by higher quality visual explanations.

Response:

Thanks for the comment. We would like to clarify that the proposed method actually achieved comparable performance with other methods. For example, as shown in **Figure 5a**, there is no significant performance difference between the proposed method and each of the other methods in the segmentation task of the brain, rib, pelvis, and bladder. Our goal is to achieve comparable performance with existing methods while largely reducing the processing time to facilitate clinical use. As shown in **Figure 5b**, the proposed method only takes less than 2 s in most segmentation tasks, while U-Net and nnU-Net take 40-200 s and 200-2000 s, respectively. It is important to note that clinicians will make final corrections to the auto-contouring results for treatment planning, so comparable performances may already meet the clinical needs.

- If the memory or patch size are the main motivations for a two-stage approach, the prior art has already shown that full volume training and testing can be achieved by model and data parallelization at scale by using a large number of GPUs. Perhaps it might be good to use the publicly available GitHub repositories to test this.

Response:

Sorry for the possible confusion. To be clear, the purpose of the two-stage approach is to perform a coarse-to-fine process, where the coarse model is to localize a minimal ROI that includes the to-be-segmented region in the original image, and then using the fine model to use this ROI as input to obtain detailed boundaries of the region. This two-stage approach can effectively exclude a large amount of irrelevant information, reduce false positives, and improve segmentation accuracy. At the same time, it helps reduce GPU memory cost and improve efficiency of segmentation. Furthermore, considering the financial cost, it may not be practical to deploy the system with the requirement of a large number of GPUs. Therefore, the lightweight RTP-Net based on a coarse-to-fine strategy is more feasible for clinical use.

- Line 66 – “accurate and robust delineation”. The paper does not talk about robustness as in resilience to imaging artefacts or population groups etc. It is probably better to rephrase it as “accurate and consistent delineation” or “reproducible”.

Response:

Thank you very much for the valuable suggestion. We have rephrased it as “accurate and consistent delineation”.

Clinical utility

Although the computation time of model is an important aspect, manual editing time or revision is equally important as the model run-time. Clinicians shouldn't be expected to spend more time correcting the model errors than annotating them from scratch. In that regard, the manuscript is not providing an extensive answer to the question whether the auto-generated contours help clinicians actually save time. The prior art has already demonstrated the clinical utility in this regard on multi-center datasets in addition to the technical advancements.

Response:

We thank the reviewer for bringing out this issue. We confirm that the total time for auto-contouring and manual editing by clinicians is much shorter than manual annotation from scratch. We have cited our previous work where our clinical collaborators reported that auto-generated contours do save time [6], described as follows:

“Compared to more than 30 min by manual delineation from scratch, the autosegmented OARs (bladder, FHs, small bowel, colon) were checked by the oncologist in the entire scan area and clinically accepted without any modification, and it took the oncologist 3 min to 8 min to modify the autosegmented CTV.”

[6] Yu, L. et al. First implementation of full-workflow automation in radiotherapy: the All-in-One solution on rectal cancer. *arXiv preprint arXiv: 2202.12009* (2022). doi: 10.48550/arXiv.2202.12009

Reviewers' Comments:

Reviewer #1:

Remarks to the Author:

I thank the editors for including the comments in the revised version of the manuscript. However, my comment 3, "The performance of the automatic segmentations should be evaluated by radiation oncologists, and additionally, the performance of re-segmentation should be addressed. " is not answered satisfactorily. The response "After 2-3 iterations, this deep learning model could achieve satisfactory segmentation performance" should be eluded to quantitatively.

Reviewer #2:

Remarks to the Author:

I would like to thank the authors for their detailed response and their efforts to address the comments to solidify the manuscript. The initial feedback was mainly intended to improve the clarity, completeness of related work, and clearer statement of the novelty of the underlying work. Therefore, I would expect the following information to be included not only in the response letter but revised manuscript for future readers' information.

- As the authors' have already indicated, CT image segmentation for image-guided radiotherapy planning has been studied extensively and it's not the first work trying to shorten treatment planning time. As you may know, there are already AI-based software commercially available used in clinical practices, which are published in various venues. Therefore, I strongly encourage the authors to extend their related work by including the references shared in the first revision, and explicitly mention the key differentiators for the clarity of future readers. (e.g., using more data, or extensive evaluation, etc.)

- As the authors included in their response, "the proposed methodology is not introducing any significant performance difference ... but reduce the processing time of segmentation algorithm", the authors should make this clearer both in the abstract and introduction to avoid any confusion.

- Most of the U-Net based solutions can process a CT volume for LINAC treatment planning within a minute or two using a decent GPU (K40-80, or any other Tesla card) that can be affordable by most of the clinics, and the same operation can be handled with CPU in the order of few minutes. I am just wondering, what difference would it make to patient's treatment time and outcome, if a model returns a segmentation in 10 seconds or 2 minutes? As you may know, in standard image-guided treatment planning, there are multiple clinical steps that take place from acquisition to delivery of the treatment including (I) a separate planning by a radiation dosimetrist, (II) treatment planning by a medical physicist, and (III) quality assurances. Is the expectation that all related staff should be waiting in the radiology department constantly for real-time planning? Could the authors include further details in the manuscript why these seconds or minute of difference matters?

REVIEWERS' COMMENTS

Reviewer #1:

I thank the editors for including the comments in the revised version of the manuscript. However, my comment 3, "The performance of the automatic segmentations should be evaluated by radiation oncologists, and additionally, the performance of re-segmentation should be addressed." is not answered satisfactorily. The response "After 2-3 iterations, this deep learning model could achieve satisfactory segmentation performance" should be eluded to quantitatively.

Response:

Sorry for the confusion. All the data were first manually delineated by radiation oncologists, serving as ground truth, and then the data were split into training and testing sets. The automatic segmentation model was built based on the training set, and the testing set was used to evaluate the model performance by comparing the model results and manual results. In this regard, the performance of automatic segmentations were evaluated by radiation oncologists.

In clinical practice, the automatic segmentation will be reviewed by radiation oncologists, and necessary modifications will be made until the level of the ground truth is reached. That is, the automatic segmentation is done only once, and the re-segmentation would be done by the radiation oncologist for final approval.

Thanks for pointing this out. The response "After 2-3 iterations, this deep learning model could achieve satisfactory segmentation performance" was aimed to convey a general concept, in which the human-in-the-loop strategy could realize iterative contouring to reduce the pressure on radiation oncologists to obtain the ground truth. The actual number of iterations varies in these 67 tasks according to the opinions of radiation oncologists. We have rephrased this description to avoid possible confusion.

Reviewer #2:

I would like to thank the authors for their detailed response and their efforts to address the comments to solidify the manuscript. The initial feedback was mainly intended to improve the clarity, completeness of related work, and clearer statement of the novelty of the underlying work. Therefore, I would expect the following information to be included not only in the response letter but revised manuscript for future readers' information.

Response:

We thank this reviewer for the valuable comments. We have revised the manuscript according to the suggestions. The corresponding responses are listed as follows.

- As the authors' have already indicated, CT image segmentation for image-guided radiotherapy planning has been studied extensively and it's not the first work trying to shorten treatment planning time. As you may know, there are already AI-based software commercially available used in clinical practices, which are published in various venues. Therefore, I strongly encourage the authors to extend their related work by including the references shared in the first revision, and explicitly mention the key differentiators for the clarity of future readers. (e.g., using more data, or extensive evaluation, etc.)

Response:

Thanks for the reviewer's valuable suggestion. We have added the following discussion.

Extracted texts from the revised manuscript:

"There are multiple AI-based software tools that are commercially available and have been used in clinical practices to standardize and accelerate the radiotherapy procedures. They include atlas-based contouring tool for automatic segmentation^{12,34-37}, and knowledge-based planning module for automatic treatment planning³⁸⁻⁴⁰. Here, we focus on exploring of AI-based automatic segmentation of target volumes and its integration into radiotherapy workflows. These AI solutions have reportedly achieved comparable performance with manual delineations in segmentation accuracy, with minor editing efforts needed^{12,35}. However, majority of the studies were only evaluated on limited organs and data with specific acquisition protocols, which affects their clinical applicability when used in different hospitals or for different target volumes. Two studies have tried to address this challenge to improve the model

generalizability^{41,42}. Nikolov et al. applied 3D U-Net to delineate 21 OARs in head and neck CT scans, and achieved expert-level performance⁴¹. The study was conducted on the training set (663 scans) and testing set (21 scans) from routine clinical practice, and validation set (39 scans) from two distinct open-source datasets. Oktay et al. incorporated the AI model into the existing radiotherapy workflow, and demonstrated that AI model could reduce contouring time while yielding clinical valid structural contours for both prostate and head-and-neck radiotherapy planning⁴². Their study involved 6 OARs for prostate cancer and 9 OARs for head and neck cancer, where experiments were conducted on a set of 519 pelvic and 242 head-and-neck CT scans acquired at eight distinct clinical sites with heterogeneous population groups and diverse image acquisition protocols. In contrast to previous works, we evaluate how RTP-Net can lead to generalized performance with extensive evaluation on 67 target volumes with varying volume sizes on a large-scale dataset of 28,581 cases (Supplementary Fig. 1). This large-scale dataset was obtained from eight distinct publicly-available datasets and one local dataset with varying acquisition settings and demographics (Supplementary Table 5). Our proposed model demonstrates performance generalizability across hospitals and target volumes, while achieving superior levels of agreement with expert contours and also time savings, which can facilitate easier deployment in clinical sites.”

“It is speculated that integrating AI-assisted delineation and AI-aided dosimetric planning into the RTP system would largely promote the efficiency of RT and reduce workload in clinical practice, such as Pinnacle³ (Philips Medical Systems, Madison, WI)⁴⁵. The proposed RTP-Net was integrated into the CT-linac system (currently being tested for clinical use approval), supporting the All-in-One radiotherapy scheme, in which the auto-contouring results (reviewed by radiation oncologists) are used for dosimetric treatment planning, to maximize the dose delivered to the tumor while minimizing the dose to the surrounding OARs. This AI-accelerated All-in-One RT workflow has two potential merits...”

- As the authors included in their response, “the proposed methodology is not introducing any significant performance difference...but reduce the processing time of segmentation algorithm”, the authors should make this clearer both in the abstract and introduction to avoid any confusion.

Response:

In this work, the proposed RTP-Net can largely reduce the processing time of contouring organs-at-risk and target tumors, while achieving comparable or superior performance with the state-of-the-art methods. We have emphasized this point in the abstract and introduction of the revised manuscript.

- Most of the U-Net based solutions can process a CT volume for LINAC treatment planning within a minute or two using a decent GPU (K40-80, or any other Tesla card) that can be affordable by most of the clinics, and the same operation can be handled with CPU in the order of few minutes. I am just wondering, what difference would it make to patient's treatment time and outcome, if a model returns a segmentation in 10 seconds or 2 minutes? As you may know, in standard image-guided treatment planning, there are multiple clinical steps that take place from acquisition to delivery of the treatment including (I) a separate planning by a radiation dosimetrist, (II) treatment planning by a medical physicist, and (III) quality assurances. Is the expectation that all related staff should be waiting in the radiology department constantly for real-time planning? Could the authors include further details in the manuscript why these seconds or minute of difference matters?

Response:

Many thanks for the valuable suggestions. Firstly, to ensure the safety and effectiveness of the radiotherapy, multiple target volumes, e.g., organs-at-risk, clinical target volume, and planning target volume, need to be accurately contoured. In one radiotherapy workflow, the number of the target volume to be contoured may reach 10, which would amplify the difference in total segmentation times, such as 100 s vs. 20 min. In this regard, the All-in-One pipeline would have the total turnaround time increasing from ~20 min to ~40 min, which would largely affect its efficiency.

Secondly, multiple clinical steps in All-in-One radiotherapy workflow need human interventions and require the presence of dedicated staff (including radiation oncologist, dosimetrist, and medical physicist) to make decision. Meanwhile, patient is on the treatment couch during the whole process. That means all related staff and patient should be waiting in the radiology department constantly from CT simulation to treatment, so there is an urgent need to improve the efficiency and save the turnaround time.

Lastly, in some clinical scenarios (especially in China), there are more patients than what a hospital could accommodate, given that medical resources (e.g., radiotherapy equipment, and professional staff) are relatively insufficient. We have added the above discussion in the revised manuscript.